# Kandinsky Conformal Prediction:
# Beyond Class- and Covariate-Conditional Coverage

**Konstantina Bairaktari** [* 1]   **Jiayun Wu** [* 2]   **Zhiwei Steven Wu** [* 2]

## Abstract

Conformal prediction is a powerful distribution-free framework for constructing prediction sets with coverage guarantees. Classical methods, such as split conformal prediction, provide *marginal coverage*, ensuring that the prediction set contains the label of a random test point with a target probability. However, these guarantees may not hold uniformly across different subpopulations, leading to disparities in coverage. Prior work has explored coverage guarantees conditioned on events related to the covariates and label of the test point. We present *Kandinsky conformal prediction*, a framework that significantly expands the scope of conditional coverage guarantees. In contrast to Mondrian conformal prediction, which restricts its coverage guarantees to disjoint groups—reminiscent of the rigid, structured grids of Piet Mondrian's art—our framework flexibly handles overlapping and fractional group memberships defined jointly on covariates and labels, reflecting the layered, intersecting forms in Wassily Kandinsky's compositions. Our algorithm unifies and extends existing methods, encompassing covariate-based group conditional, class conditional, and Mondrian conformal prediction as special cases, while achieving a minimax-optimal high-probability conditional coverage bound. Finally, we demonstrate the practicality of our approach through empirical evaluation on real-world datasets.

---

[*]Equal contribution [1]Khoury College of Computer Sciences, Northeastern University, Boston, MA, USA. [2]School of Computer Science, Carnegie Mellon University, Pittsburgh, PA, USA. Correspondence to: Konstantina Bairaktari <bairaktari.k@northeastern.edu>, Jiayun Wu <jiayunw@andrew.cmu.edu>, Zhiwei Steven Wu <zstevenwu@cmu.edu>.

*Proceedings of the 42$^{nd}$ International Conference on Machine Learning*, Vancouver, Canada. PMLR 267, 2025. Copyright 2025 by the author(s).

## 1. Introduction

Conformal prediction (Vovk et al., 2005; Lei & Wasserman, 2014; Papadopoulos et al., 2002b) is a framework for constructing prediction sets with formal coverage guarantees. Given a predictor $f : \mathcal{X} \rightarrow \mathcal{Y}$ and $n$ samples $\{(X_i, Y_i)\}_{i \in [n]}$ from a distribution $\mathcal{D}$, the goal is to build a prediction set function $\mathcal{C}$ that, for a given covariate vector $X_{n+1}$, outputs a subset of $\mathcal{Y}$ guaranteed to include the true label $Y$ with a target probability. Ideally, the size of $\mathcal{C}(X_{n+1})$ reflects the uncertainty in $f(X_{n+1})$.

A common coverage guarantee, termed as *training conditional validity* (Vovk, 2012), requires that for any $f$ the training algorithm returns a (potentially randomized) prediction set function $\mathcal{C}$ such that, with probability $1 - \delta$ over the draws of the calibration dataset $\{(X_i, Y_i)\}_{i \in [n]}$, the following holds:

$$\mathbb{P}\left[Y_{n+1} \in \mathcal{C}(X_{n+1}) \mid \{(X_i, Y_i)\}_{i \in [n]}\right] \approx 1 - \alpha, \quad (1)$$

for some $\alpha$ and $\delta$ in $(0, 1)$. This guarantee ensures coverage *on average* over random test samples $(X_{n+1}, Y_{n+1})$ drawn from $\mathcal{D}$. A weaker guarantee, called the *marginal coverage guarantee*, requires that $\mathbb{P}[Y_{n+1} \in \mathcal{C}(X_{n+1})] \approx 1 - \alpha$, where the probability is taken over all $n + 1$ points (Vovk et al., 2005; Gibbs et al., 2023). However, a limitation of such average coverage guarantees is that the prediction sets might undercover certain values of $(X_{n+1}, Y_{n+1})$ while overcovering others. A potential solution is to require $(1-\alpha)$-coverage conditioning on the value of $(X_{n+1}, Y_{n+1})$. Unfortunately, we generally cannot obtain non-trivial prediction sets that satisfy such pointwise coverage (Vovk, 2012; Lei & Wasserman, 2014; Barber et al., 2021).

However, various types of conditional coverage guarantees can still be achieved. For instance, Mondrian conformal prediction (Vovk et al., 2003) ensures the $(1 - \alpha)$-coverage guarantees over a disjoint set of subgroups $\mathcal{G}$ defined over both the covariate $X_{n+1}$ and the label $Y_{n+1}$. The method is named after Piet Mondrian, as its partitioning the space $\mathcal{X} \times \mathcal{Y}$ into a finite set of non-overlapping groups $\mathcal{G}$ mirrors Mondrian's iconic compositions of disjoint rectangles (see Figure 1). Class-conditional conformal prediction (for classification) (Löfström et al., 2015; Ding et al., 2023) can be viewed as a special case of Mondrian conformal

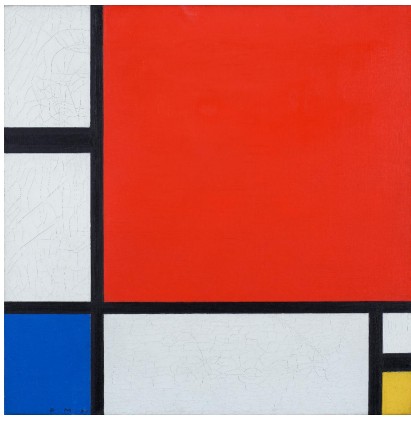

Figure 1: Composition with Red, Blue and Yellow, Piet Mondrian, 1930, Kunsthaus Zurich

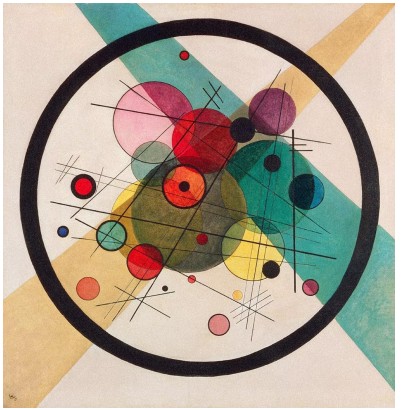

Figure 2: Circles in a Circle, Wassily Kandinsky, 1923, Philadelphia Museum of Art

prediction, where each label defines a distinct group.

Yet, the disjoint group assumption in Mondrian conformal prediction has notable limitations, as real-world subpopulations of interest often overlap. This is particularly relevant in algorithmic fairness, where protected subgroups–typically defined by combinations of demographic attributes such as race and gender—naturally intersect (Kearns et al., 2018; Hébert-Johnson et al., 2018). An alternative line of research addresses this challenge by developing conditional coverage guarantees for overlapping groups (Jung et al., 2023; Gibbs et al., 2023). However, both of their results restrict the group functions to depend solely on the covariate vector $X_{n+1}$.

In this paper, we expand the scope of conditional coverage guarantees by achieving the best of both worlds from these two lines of prior work. Building on Gibbs et al. (2023); Jung et al. (2023), we provide coverage guarantees for overlapping groups while, in the spirit of Mondrian conformal prediction (Vovk et al., 2003), allowing grouping functions to depend jointly on both the covariates $X_{n+1}$ and the label $Y_{n+1}$. More generally, our guarantees hold for fractional groups, where membership is defined as a probabilistic function over $X_{n+1}$ and $Y_{n+1}$. This added flexibility enables the modeling of subgroups based on protected attributes that are not explicitly observed but can be inferred through covariates and labels (Romano et al., 2020a). We name our method *Kandinsky Conformal Prediction*[1], drawing inspiration from Wassily Kandinsky's compositions of overlapping geometric forms (see Figure 2), as we provide conditional coverage guarantees for groups that are flexible, overlapping, and probabilistic.

---

[1]We note that the name Kandinsky Conformal Prediction coincides with that of an unrelated conformal prediction method for image segmentation (Brunekreef et al., 2024), which balances pixel-level and image-level calibration. Our technique instead focuses on conditional guarantees of general conformal prediction.

Similar to Gibbs et al. (2023), our conditional guarantee can be leveraged to obtain valid coverage guarantees under distribution shift, provided that the density ratio between the test and calibration distributions is captured by one of the group functions we consider. While Gibbs et al. (2023) is limited to handling covariate shift—since their group functions depend solely on $X$ —our approach extends to a broader class of distribution shifts affecting both $X$ and $Y$.

Our algorithm is computationally efficient, which relies on solving a linear quantile regression over the vector space spanned by a set of group functions $\mathcal{G}$. Similar to Jung et al. (2023), it preserves the key advantage of requiring only a single quantile regression model that can be applied to all test examples, rather than needing to solve a separate regression for each instance as in Gibbs et al. (2023). We show that our method achieves a high-probability conditional coverage error rate of $\mathcal{O}(n_{\mathcal{G}}^{-1/2})$, where $n_{\mathcal{G}}$ is the number of samples per group on average, matching the minimax-optimal rate (Areces et al., 2024). In the special case where group functions only depend on the covariates, our bound significantly improves the $\widetilde{\mathcal{O}}(n_{\mathcal{G}}^{-1/4})$ coverage error bound in Roth (2022); Jung et al. (2023) in its dependence on $n_{\mathcal{G}}$, and sharpens the dependence on $|\mathcal{G}|$ when compared to the $\mathcal{O}(n_{\mathcal{G}}^{-1/2}(1 + |\mathcal{G}|^{1/2} n_{\mathcal{G}}^{-1/2}))$ bound in Areces et al. (2024).

To complement our main algorithm, which ensures training-conditional validity, we also provide a *test-time inference* algorithm that obtains expected marginal coverage that also takes randomness over the draws on calibration data. By extending Gibbs et al. (2023)'s approach to accommodate group functions defined over both covariates and labels, we obtain an expected coverage error bound of $\mathcal{O}(n_{\mathcal{G}}^{-1})$. However, this result comes at the cost of computational efficiency, as the algorithm needs to perform test-time inference that solves an instance of quantile

regression for every new test example.

Our contribution can be summarized as follows.

1. We study the most general formulation of group-conditional coverage guarantees in conformal prediction, handling overlapping groups and fractional group memberships that are defined jointly by covariates and labels. This conditional guarantee can be used to obtain valid coverage under general subpopulation distribution shifts.

2. We propose Kandinsky Conformal Prediction, a method that is both computationally efficient and statistically optimal. Computationally, it requires solving a single quantile regression over the vector space spanned by group functions. Statistically, it attains the minimax-optimal group-conditional coverage error with finite samples.

3. We evaluate our algorithm on real-world tasks, including income prediction and toxic comment detection. Our results show that Kandinsky Conformal Prediction consistently achieves the best-calibrated conditional coverage while scaling effectively with the number of groups.

## 1.1. Related Work

The literature on conformal prediction is vast, covering a range of coverage guarantees, methods, and applications. Several textbooks and surveys provide in-depth discussions (Vovk et al., 2005; Shafer & Vovk, 2008; Balasubramanian et al., 2014; Angelopoulos et al., 2024). A classical method is split conformal prediction (Papadopoulos et al., 2002a; Lei et al., 2018), which ensures marginal coverage. A major advantage of conformal prediction is that the coverage guarantees hold regardless of the choice of non-conformity score functions. Meanwhile, another line of research explores specialized non-conformity scores for specific tasks, such as regression (Lei et al., 2013; Romano et al., 2019; Izbicki et al., 2020) and classification (Sadinle et al., 2019; Angelopoulos et al., 2020; Romano et al., 2020b).

Our work is closely related to the growing body of work on conformal prediction that establishes coverage guarantees conditioned on events involving the test example $(X_{n+1}, Y_{n+1})$. A fundamental class of such events corresponds to disjoint subgroups. For instance, Mondrian conformal prediction (Vovk et al., 2003) partitions the space $\mathcal{X} \times \mathcal{Y}$ into disjoint regions, encompassing class-conditional (Löfström et al., 2015; Ding et al., 2023) and sensitive-covariate-conditional (Romano et al., 2020a) conformal prediction as special cases.

We introduce Kandinsky Conformal Prediction as an extension of the Mondrian approach, providing coverage guarantees for overlapping group structures. Prior work on overlapping group-conditional coverage primarily considers groups defined solely by covariates $X_{n+1}$. Barber et al. (2021) addresses this by assigning the most conservative prediction set to points in multiple groups. Our algorithm builds on the quantile regression techniques of Jung et al. (2023) and Gibbs et al. (2023), which compute per-example thresholds on non-conformity scores. Blot et al. (2024) extends Gibbs et al. (2023)'s technique to include group membership functions depending on both the label and the covariate, and they establish a one-sided marginal coverage bound. We further provide two-sided training conditional bounds with a different algorithm. Additionally, Areces et al. (2024) establishes a lower bound on group-conditional coverage error rates. Since we generalize to group functions that depend on both covariates and labels, their bound applies to our setting and shows that our error rate is minimax-optimal.

Thematically, our work is related to a long line of work in multi-group fairness (Kearns et al., 2018; Hébert-Johnson et al., 2018; Kearns et al., 2019). In particular, our algorithm can be seen as predicting a target quantile conditioned on covariates and labels while satisfying the *multiaccuracy* criterion (Kim et al., 2019; Roth, 2022).

Finally, our work advances the study of conformal prediction under distribution shifts between calibration and test data. Existing research largely focuses on specific cases, such as covariate shift (Tibshirani et al., 2019; Qiu et al., 2023; Yang et al., 2024; Pournaderi & Xiang, 2024) or label shift (Podkopaev & Ramdas, 2021). Plassier et al. (2024) considers general shifts on both the covariate and label distributions. However, all of these approaches are designed for a single target distribution. By introducing a more general class of group functions that depend jointly on covariates and labels, we establish coverage guarantees that remain valid under broader distribution shifts over $\mathcal{X} \times \mathcal{Y}$, including multiple potential target distributions.

## 2. Preliminaries

We consider prediction tasks over $\mathcal{X} \times \mathcal{Y}$, where $\mathcal{X}$ represents the covariate domain and $\mathcal{Y}$ represents the label domain. $\mathcal{Y}$ can be either finite for classification tasks or infinite for regression tasks. Unless stated otherwise, we assume that the observed data is drawn from a distribution $\mathcal{D}$, defined over $\mathcal{X} \times \mathcal{Y}$. We consider potentially randomized prediction set functions, denoted by $\mathcal{C}(X; \varepsilon)$, where the randomness is introduced by a random variable $\varepsilon$ that is independent of $(X, Y)$. We further assume that the randomness of the conformal prediction algorithm depends on $n$ independent random variables $\varepsilon_i$ that are independent of the dataset $\{(X_i, Y_i)\}_{i \in [n]}$.

Let $(1 - \alpha)$ be the target coverage level, where $\alpha \in (0, 1)$. We can now restate the definition of training conditional validity to account for the randomness of the algorithm and the prediction set function. Specifically, this requires that for any $f$ the training algorithm returns a (potentially randomized) prediction set function $\mathcal{C}$ such that, with probability $1 - \delta$ over the draws of the calibration dataset $\{(X_i, Y_i)\}_{i \in [n]}$ and the randomness of the conformal prediction algorithm, captured by $\{\varepsilon_i\}_{i \in [n]}$, the following holds:

$$\mathbb{P}\left[Y_{n+1} \in \mathcal{C}(X_{n+1}; \varepsilon_{n+1}) \mid \{(X_i, Y_i, \varepsilon_i)\}_{i \in [n]}\right] \approx 1 - \alpha, \tag{2}$$

for some $\delta$ in $(0, 1)$.

We use weight functions $w : \mathcal{X} \times \mathcal{Y} \to \mathbb{R}$ to model our conditional coverage guarantees. For any weight function $w$, we measure the error of a fixed prediction set function $\mathcal{C}$ by its *weighted coverage deviation*:

$$\mathrm{wCD}(\mathcal{C}, \alpha, w) := \mathbb{E}[w(X, Y)(\mathbb{1}\{Y \in \mathcal{C}(X; \varepsilon)\} - (1 - \alpha))],$$

where the expectation is taken over the test point $(X, Y)$, drawn from $\mathcal{D}$, and $\varepsilon$, the internal randomness of $\mathcal{C}$.

Many conformal prediction methods require calculating quantiles of a distribution.

**Definition 2.1** (Quantile). Given a distribution $P$ defined on $\mathbb{R}$, for any $\tau \in [0, 1]$ the $\tau$-quantile of the distribution $P$, denoted $q$, is defined as

$$q = \inf\{x \in \mathbb{R} : \mathbb{P}_{X \sim P}[X \leq x] \geq \tau\}.$$

Our more general framework performs quantile regression by solving a pinball loss minimization problem. By Lemma 2.3, in the special case where we compute the largest value in $\mathbb{R}$ that minimizes the sum of pinball losses over a set of points, we obtain a $(1 - \alpha)$ quantile of the set.

**Definition 2.2** (Pinball Loss). For a given target quantile $1 - \alpha$, where $\alpha \in [0, 1]$, predicted value $\theta \in \mathbb{R}$ and score $s \in \mathbb{R}$, the pinball loss is defined as

$$\ell_\alpha(\theta, s) := \begin{cases} (1 - \alpha)(s - \theta), & \text{if } s \geq \theta, \\ \alpha(\theta - s), & \text{if } s < \theta. \end{cases}$$

**Lemma 2.3** ((Koenker & Bassett, 1978)). *Let $\alpha$ be a parameter in $(0, 1)$, and let $\{s_i\}_{i \in [n]}$ be a set of $n$ points, where for all $i \in [n]$, $s_i \in \mathbb{R}$. Then, the largest $\theta^* \in \mathbb{R}$ such that*

$$\theta^* \in \arg\min_{\theta \in \mathbb{R}} \sum_{i \in [n]} \ell_\alpha(\theta, s_i)$$

*is a $(1 - \alpha)$-quantile of $\{s_i\}_{i \in [n]}$.*

## 3. Kandinsky Conformal Prediction

In this section, we describe the components of the *Kandinsky conformal prediction* framework. We will first present our method by extending the conditional conformal prediction algorithm by Jung et al. (2023) to work with classes of finite-dimensional weight functions of the form $\mathcal{W} = \{\Phi(\cdot)^T \beta : \beta \in \mathbb{R}^d\}$, where $\Phi : \mathcal{X} \times \mathcal{Y} \to \mathbb{R}$ is a basis that is defined according to the desired coverage guarantee. After presenting our main result, we will demonstrate how this conformal prediction method can be tailored to different applications. The detailed proofs of our results from this section are in Appendix A.1.

Given a coverage parameter $\alpha \in (0, 1)$, a predictor $f : \mathcal{X} \to \mathcal{Y}$, and a calibration dataset of $n$ examples $D = \{(X_i, Y_i)\}_{i \in [n]}$, drawn independently from a distribution $\mathcal{D}$, the objective of Kandinsky conformal prediction is to construct a (possibly randomized) prediction set function $\mathcal{C}(\cdot)$ such that, with high probability over the calibration dataset $D$ and the randomness of the method that constructs $\mathcal{C}$, for every $w \in \mathcal{W}$

$$\mathrm{wCD}(\mathcal{C}, \alpha, w) \approx 0.$$

Given the predictor $f$, we use a non-conformity score function $S : \mathcal{X} \times \mathcal{Y} \to \mathbb{R}$ to measure how close the prediction $f(x)$ is to the label $y$ for any data point $(x, y) \in \mathcal{X} \times \mathcal{Y}$. As is common in conformal prediction, our method ensures the desirable coverage for any score function $S$. However, in our more general framework, we allow for the use of a randomized score function to break potential ties in quantile regression, while keeping the assumptions about distribution $\mathcal{D}$ minimal. We implement this through a randomized score function $\widetilde{S} : \mathcal{X} \times \mathcal{Y} \times \mathbb{R} \to \mathbb{R}$ that takes as input the covariates, labels and some random noise $\varepsilon \in \mathbb{R}$.

Our method consists of two steps: one is performed during training, and the other at test time. Similar to Jung et al. (2023), the first step is a quantile regression on the scores of the $n$ data points in the calibration dataset, $\{S(X_i, Y_i)\}_{i \in [n]}$. For randomized score functions, we work with the set of randomized scores $\{\widetilde{S}(X_i, Y_i, \varepsilon_i)\}_{i \in [n]}$, where for every point $i$, $\varepsilon_i$ is drawn independently from a distribution $\mathcal{D}_{\mathrm{rn}}$. In Algorithm 1, we compute a $(1 - \alpha)$-"quantile function" $\widehat{q}$. This $\widehat{q}$ is a weight function from $\mathcal{W}$ that minimizes the average pinball loss of the $n$ (randomized) scores with parameter $\alpha$.

At test time, we run the second step that constructs the prediction set for a given point with covariates $X_{n+1}$. For a fixed estimated quantile function $\widehat{q}$, the prediction set produced by Algorithm 2 includes all labels $y \in \mathcal{Y}$ whose (randomized) score (with noise $\varepsilon_{n+1}$ drawn from $\mathcal{D}_{\mathrm{rn}}$) is at most $\widehat{q}(X_{n+1}, y)$. For more details about the computation of Algorithm 2 see Appendix A.3.

---

**Algorithm 1** Quantile Regression of Kandinsky CP

---

**Input:** $\{(X_i, Y_i)\}_{i \in [n]}, \mathcal{W}$, and $\widetilde{S}$

1: For all $i \in [n]$ draw $\varepsilon_i$ independently from $\mathcal{D}_{\text{rn}}$.
2: Find

$$\widehat{q} \in \arg\min_{w \in \mathcal{W}} \frac{1}{n} \sum_{i=1}^{n} \ell_\alpha(w(X_i, Y_i), \widetilde{S}(X_i, Y_i, \varepsilon_i)).$$

**Return:** Quantile function $\widehat{q}$

---

**Algorithm 2** Prediction Set Function of Kandinsky CP

---

**Input:** $X_{n+1}, \widehat{q}$, and $\widetilde{S}$

1: Draw $\varepsilon_{n+1}$ independently from $\mathcal{D}_{\text{rn}}$.
2: Set $\mathcal{C}(X_{n+1}; \varepsilon_{n+1}) =$

$$\left\{ y : \widetilde{S}(X_{n+1}, y, \varepsilon_{n+1}) \leq \widehat{q}(X_{n+1}, y) \right\}.$$

**Return:** $\mathcal{C}(X_{n+1}; \varepsilon_{n+1})$

---

Our main result is stated in Theorem 3.1, where we prove that, under the assumption that the distribution of the randomized score $\widetilde{S}$ conditioned on the value of the basis $\Phi$ is continuous, the weighted coverage deviation of $\mathcal{C}$ converges to zero with high probability at a rate of $\mathcal{O}(\sqrt{d/n} + d/n)$, where $d$ is the dimension of $\Phi$. When the distribution of $S(X, Y) \mid \Phi(X, Y)$ is continuous, this theorem holds for the deterministic score function $S$, since setting $\widetilde{S}(x, y, \varepsilon) = S(x, y)$ satisfies the assumptions.

**Theorem 3.1.** *Let parameters $\alpha, \delta \in (0, 1)$, and $\mathcal{W} = \{\Phi(\cdot)^T \beta : \beta \in \mathbb{R}^d\}$ denote a class of linear weight functions over a bounded basis $\Phi : \mathcal{X} \times \mathcal{Y} \to \mathbb{R}^d$. Without loss of generality, assume $\|\Phi(x, y)\|_\infty \leq 1$ for all $x, y$. Assume that the data $\{(X_i, Y_i)\}_{i \in [n]}$ are drawn i.i.d. from a distribution $\mathcal{D}$, $\{\varepsilon_i\}_{i \in [n]}$ are drawn i.i.d. from a distribution $\mathcal{D}_{\text{rn}}$, independently from the dataset, and the distribution of $\widetilde{S}(X, Y, \varepsilon) \mid \Phi(X, Y)$ is continuous. There exists an absolute constant $C$ such that, with probability at least $1 - \delta$ over the randomness of the calibration dataset $\{(X_i, Y_i)\}_{i \in [n]}$ and the noise $\{\varepsilon_i\}_{i \in [n]}$, the (randomized) prediction set $\mathcal{C}$ given by Algorithms 1 and 2 satisfies, for every $w_\beta = \Phi(\cdot)^T \beta$,*

$$|\text{wCD}(\mathcal{C}, \alpha, w_\beta)| \leq$$
$$\|\beta\|_1 \left( C\sqrt{\frac{d}{n}} + \frac{d}{n} + \max\{\alpha, 1 - \alpha\} \sqrt{\frac{2\ln(4d/\delta)}{n}} \right).$$

As a sketch of the proof, we first establish a connection between the subgradient of the empirical pinball loss and the empirical weighted coverage. We then prove a concentration result for the weighted coverage. In contrast, Roth (2022);

Jung et al. (2023) derive concentration results directly for the pinball loss, which leads to a slower convergence rate of $\mathcal{O}((d \log n/n)^{1/4})$.

As a corollary of Theorem 3.1, we derive a bound on the expected coverage deviation of $\mathcal{C}$, with the expectation taken over the randomness of the calibration dataset and the noise in the randomized scores.

**Corollary 3.2.** *Let $\alpha, \delta, \mathcal{W}, w_\beta$ be specified as in Theorem 3.1 and consider the same assumptions on $\{(X_i, Y_i, \varepsilon_i)\}_{i \in [n]}$, $\widetilde{S}$, and $\Phi$ in Theorem 3.1. Then, there exists an absolute constant $C$ such that the (randomized) prediction set $\mathcal{C}$, given by Algorithms 1 and 2, satisfies*

$$\mathbb{E}_{D, E} \left[ \sup_{w_\beta \in \mathcal{W}} \frac{\text{wCD}(\mathcal{C}, \alpha, w_\beta)}{\|\beta\|_1} \right] \leq C\sqrt{\frac{d}{n}} + \frac{d}{n},$$

*where $D$ is the calibration dataset $\{(X_i, Y_i)\}_{i \in [n]}$ and $E$ is the corresponding noise $\{\varepsilon_i\}_{i \in [n]}$.*

**Weight Functions Defined on the Covariates** Areces et al. (2024) show that when the covariate-based weight class $\mathcal{W}$ contains a class of binary functions with VC dimension $d$, equal to the dimension of the basis $\Phi$, then the convergence rate of the expected coverage deviation is at least $O(\sqrt{d\alpha(1 - \alpha)/n})$. This means that for $n > d$, the rate in Corollary 3.2 is asymptotically optimal. In Appendix A.2 we compare our upper bounds with prior work.

### 3.1. Applications

Kandinsky conformal prediction is a general framework that can be specialized to obtain several types of weighted or conditional conformal prediction. The choice of the appropriate basis $\Phi$ and, consequently, the weight function class $\mathcal{W}$ depends on the desired coverage guarantee. In this subsection, we explore how Kandinsky conformal prediction can be applied to several scenarios, including group-conditional conformal prediction, Mondrian conformal prediction, group-conditional conformal prediction with fractional group membership, and conformal prediction under distribution shift.

**Group-Conditional Conformal Prediction** Suppose we want to achieve the target coverage $(1 - \alpha)$ for every group of points $G \subset \mathcal{X} \times \mathcal{Y}$ within a finite set of potentially overlapping groups $\mathcal{G}$. Let $\pi_G(\mathcal{C})$ be the probability that the true label is included in a fixed prediction set $\mathcal{C}$, conditioned on the datapoint $(X_{n+1}, Y_{n+1})$ belonging to some group $G$. Formally, with the calibration dataset $D = \{(X_i, Y_i)\}_{i \in [n]}$ and the corresponding noise $E = \{\varepsilon_i\}_{i \in [n]}$, let

$$\pi_G(\mathcal{C}) :=$$
$$\mathbb{P}[Y_{n+1} \in \mathcal{C}(X_{n+1}; \varepsilon_{n+1}) \mid D, E, (X_{n+1}, Y_{n+1}) \in G].$$

We want to ensure that for every $G \in \mathcal{G}$, $\pi_G(\mathcal{C}) \approx 1 - \alpha$. Here we can define $\Phi$ to be a $|\mathcal{G}|$-dimensional vector that has an entry $\mathbb{1}\{(x, y) \in G\}$ for every $G \in \mathcal{G}$. Corollary 3.3 provides the rate at which $\pi_G(\mathcal{C})$ converges to $(1 - \alpha)$, when $\mathcal{C}$ is constructed by Algorithms 1 and 2. For conciseness, we define $\pi_G := \mathbb{P}[(X_{n+1}, Y_{n+1}) \in G]$.

**Corollary 3.3.** *Let parameters $\alpha, \delta \in (0, 1)$, and $\mathcal{W} = \{\sum_{G \in \mathcal{G}} \beta_G \mathbb{1}\{(x, y) \in G\} : \beta_G \in \mathbb{R}, \forall G \in \mathcal{G}\}$. Assume $\{(X_i, Y_i, \varepsilon_i)\}_{i \in [n+1]}$ are i.i.d. samples. Under the same assumptions on $\widetilde{S}$ and $\Phi$ in Theorem 3.1, there exists an absolute constant $C$ such that, with probability at least $1 - \delta$ over the randomness of the calibration dataset $\{(X_i, Y_i)\}_{i \in [n]}$ and the noise $\{\varepsilon_i\}_{i \in [n]}$, the (randomized) prediction set $\mathcal{C}$ given by Algorithms 1 and 2 satisfies, for all $G \in \mathcal{G}$,*

$$|\pi_G(\mathcal{C}) - (1 - \alpha)| \leq$$
$$\frac{1}{\pi_G} \left( C\sqrt{\frac{|\mathcal{G}|}{n}} + \frac{|\mathcal{G}|}{n} + \max\{\alpha, 1 - \alpha\}\sqrt{\frac{2\ln(4|\mathcal{G}|/\delta)}{n}} \right).$$

**Mondrian Conformal Prediction** A special case of group-conditional conformal prediction is Mondrian conformal prediction, where groups in $\mathcal{G}$ are disjoint and form a partition of the domain $\mathcal{X} \times \mathcal{Y}$. Since each $(X_i, Y_i)$ in the calibration dataset belongs to exactly one group in $\mathcal{G}$, we can verify that Algorithm 1 computes a value $\beta_G \in \mathbb{R}$ for every group $G$ independently. By Lemma 2.3, for every $G \in \mathcal{G}$, if $\beta_G$ is the largest value that minimized the pinball loss, then it is a $(1 - \alpha)$-quantile of the scores of the calibration datapoints that belong to $G$. As a result, for a test point $X_{n+1}$, we construct the prediction set $\mathcal{C}(X_{n+1})$ by including every $y$ where the score $\widetilde{S}(X_{n+1}, y, \varepsilon)$ is at most the quantile $\beta_G$ corresponding to the group $G$ that contains $(X_{n+1}, y)$. This method is the same as the one presented in (Vovk et al., 2003), with the difference that in that method, the quantiles are slightly adjusted to achieve the expected marginal coverage guarantee through exchangeability.

**Fractional Group Membership** In some cases, we are concerned with the conditional coverage in groups defined by unobserved attributes. Let $\mathcal{Z}$ be the domain of these unobserved attributes, and let $\mathcal{D}'$, defined over $\mathcal{X} \times \mathcal{Y} \times \mathcal{Z}$, be the joint distribution of the covariates, the label, and the unobserved attributes. In this setting, $\mathcal{G}$ is a set of groups defined as subsets of $\mathcal{Z}$. Let

$$\pi_G^Z(\mathcal{C}) := \mathbb{P}[Y_{n+1} \in \mathcal{C}(X_{n+1}; \varepsilon_{n+1}) \mid D, E, Z_{n+1} \in G].$$

Then, we want to ensure that $\pi_G^Z(\mathcal{C}) \approx 1 - \alpha$ for every $G \in \mathcal{G}$. To achieve this, we set the basis of the weight function class as the probability of $Z \in G$ given a statistic $\phi(X, Y)$. For every $G \in \mathcal{G}$,

$$\Phi_G(x, y) = \mathbb{P}[Z \in G \mid \phi(X, Y) = \phi(x, y)]. \quad (3)$$

The simplest example of the statistic $\phi$ is $\phi_{XY}(X, Y) = (X, Y)$. In practice, if a pretrained $\Phi$ is not available, we may use the protected attributes $Z$ in calibration data to estimate the probabilities in $\Phi$. In some cases, we may only have indirect access to the covariates through the pretrained predictor $f(X)$. Then, we can construct the statistic as $\phi_{FY}(X, Y) = (f(X), Y)$. In general, Corollary 3.4 shows that our algorithm ensures group conditional coverage if $\phi$ is a sufficient statistic for the score function, which is satisfied by both $\phi_{XY}, \phi_{FY}$.

For our theoretical result, we assume that we already know these conditional probabilities, $\Phi_G(x, y)$, for all $G \in \mathcal{G}$, and that the calibration and the test data do not include unobserved attributes. Corollary 3.4 shows that by constructing $\mathcal{C}$ according to Algorithms 1 and 2, the coverage converges to $(1 - \alpha)$ at an $O(\sqrt{|\mathcal{G}|/n} + |\mathcal{G}|/n)$ rate. For brevity, we define $\pi_G^Z := \mathbb{P}[Z_{n+1} \in G]$.

**Corollary 3.4.** *Let $\alpha, \delta \in (0, 1)$, $\phi(X, Y)$ be a sufficient statistic for $\widetilde{S}$, such that $\widetilde{S}$ is conditionally independent of $X, Y$ given $\phi$, and $\mathcal{W} = \{\sum_{G \in \mathcal{G}} \beta_G \Phi_G : \beta_G \in \mathbb{R}, \forall G \in \mathcal{G}\}$. Assume $\{(X_i, Y_i, \varepsilon_i)\}_{i \in [n+1]}$ are i.i.d. samples. Under the same assumptions on $\widetilde{S}$ and $\Phi$ in Theorem 3.1, there exists an absolute constant $C$ such that, with probability at least $1 - \delta$ over the randomness of the calibration dataset $\{(X_i, Y_i)\}_{i \in [n]}$ and the noise $\{\varepsilon_i\}_{i \in [n]}$, the (randomized) prediction set $\mathcal{C}$ given by Algorithms 1 and 2 satisfies, for all $G \in \mathcal{G}$,*

$$\left|\pi_G^Z(\mathcal{C}) - (1 - \alpha)\right| \leq$$
$$\frac{1}{\pi_G^Z} \left( C\sqrt{\frac{|\mathcal{G}|}{n}} + \frac{|\mathcal{G}|}{n} + \max\{\alpha, 1 - \alpha\}\sqrt{\frac{2\ln(4|\mathcal{G}|/\delta)}{n}} \right).$$

**Distribution Shifts** Kandinsky conformal prediction can also ensure the target coverage under a class of distribution shifts, where the joint distribution over both the covariates and the labels can change. In this context, we assume that the calibration dataset $\{(X_i, Y_i)\}_{i \in [n]}$ is sampled i.i.d. from a source distribution $\mathcal{D}$, while the test example $(X_{n+1}, Y_{n+1})$ may be drawn independently from another distribution $\mathcal{D}_T$. The objective is to construct a prediction set that satisfies

$$\mathbb{P}_{\mathcal{D}_T}[Y_{n+1} \in \mathcal{C}(X_{n+1}; \varepsilon_{n+1}) \mid \{X_i, Y_i, \varepsilon_i\}_{i \in [n]}] \approx 1 - \alpha,$$

for a set of test distributions $\mathcal{D}_T$. Given a weight function class $\mathcal{W}$, Corollary 3.5 states that we can achieve the desired coverage for all distribution shifts $\mathcal{D}_T$ with its Radon-Nikodym derivative $\frac{d\mathbb{P}_{\mathcal{D}_T}}{d\mathbb{P}_{\mathcal{D}}}(x, y)$ contained in $\mathcal{W}$.

**Corollary 3.5.** *Let $\alpha$ and $\delta$ be parameters in $(0, 1)$, and $\mathcal{W} = \{\Phi(\cdot)^T \beta : \beta \in \mathbb{R}^d\}$ denote a class of linear weight functions over a bounded basis $\Phi : \mathcal{X} \times \mathcal{Y} \to \mathbb{R}^d$. Under the same assumptions on $\{(X_i, Y_i, \varepsilon_i)\}_{i \in [n]}$, $\widetilde{S}$, and $\Phi$ in*

*Theorem 3.1, there exists an absolute constant $C$ such that, for every distribution $\mathcal{D}_T$ such that $\frac{d\mathbb{P}_{\mathcal{D}_T}}{d\mathbb{P}_{\mathcal{D}}} \in \mathcal{W}$ and $\left|\frac{d\mathbb{P}_{\mathcal{D}_T}}{d\mathbb{P}_{\mathcal{D}}}(x,y)\right| \leq B$ for any $x \in \mathcal{X}$ and $y \in \mathcal{Y}$, with probability at least $1 - \delta$ over the randomness of the calibration dataset $\{(X_i, Y_i)\}_{i\in[n]}$ and the noise $\{\varepsilon_i\}_{i\in[n]}$, the (randomized) prediction set $\mathcal{C}$, given by Algorithms 1 and 2, satisfies*

$$\left|\mathbb{P}[Y_{n+1} \in \mathcal{C}(X_{n+1}; \varepsilon_{n+1}) \mid \{X_i, Y_i, \varepsilon_i\}_{i\in[n]}] - (1-\alpha)\right|$$
$$\leq B\left(C\sqrt{\frac{d}{n}} + \frac{d}{n} + \max\{\alpha, 1-\alpha\}\sqrt{\frac{2\ln(4/\delta)}{n}}\right),$$

*where $(X_{n+1}, Y_{n+1})$ are drawn independently from the distribution $\mathcal{D}_T$, and $\varepsilon_{n+1}$ are drawn independently from the distribution $\mathcal{D}_{rn}$.*

## 4. Test-time Quantile Regression

In this section, we explore an alternative quantile regression-based algorithm that extends the method proposed in (Gibbs et al., 2023) to weight function classes over $\mathcal{X} \times \mathcal{Y}$. This method, detailed in Algorithm 3, is more complicated than Algorithms 1, 2 and performs both steps during test time.

Specifically, given the covariates of a test point $X_{n+1}$, Algorithm 3 computes a separate quantile function $\widehat{q}_y$ for each label $y \in \mathcal{Y}$ by minimizing the average pinball loss over the scores of both $\{(X_i, Y_i)\}_{i\in[n]}$ and $(X_{n+1}, y)$. Since $X_{n+1}$ is required for the quantile regression, this step is performed at test time. The prediction set $\mathcal{C}$ produced by Algorithm 3 includes all labels $y \in \mathcal{Y}$ where the score is at most the label specific quantile, evaluated at $(X_{n+1}, y)$.

---

**Algorithm 3** Test-time Quantile Regression

**Input:** $X_{n+1}$, $\{(X_i, Y_i)\}_{i\in[n]}$, $\mathcal{W}$, and $\widetilde{S}$

1: For all $i \in [n+1]$, draw $\varepsilon_i$ independently from $\mathcal{D}_{rn}$
2: Set $\mathcal{C}(X_{n+1}; \varepsilon_{n+1}) =$

$$\{y : \widetilde{S}(X_{n+1}, y, \varepsilon_{n+1}) \leq \widehat{q}_y(X_{n+1}, y)\},$$

where

$$\widehat{q}_y \in \arg\min_{w\in\mathcal{W}} \frac{1}{n+1}\sum_{i=1}^{n} \ell_\alpha(w(X_i, Y_i), \widetilde{S}(X_i, Y_i, \varepsilon_i))$$
$$+ \frac{1}{n+1}\ell_\alpha(w(X_{n+1}, y), \widetilde{S}(X_{n+1}, y, \varepsilon_{n+1}))).$$

**Return:** $\mathcal{C}(X_{n+1}; \varepsilon_{n+1})$

---

For the weight functions $\mathcal{W} = \{\Phi(\cdot)^T\beta : \beta \in \mathbb{R}^d\}$, where $\Phi : \mathcal{X} \times \mathcal{Y} \to \mathbb{R}^d$, under the same assumptions about the data and the distribution of $\widetilde{S} \mid \Phi$ in Theorem 3.1,

the expected weighted coverage deviation is bounded by $O(d/n)$.

**Theorem 4.1.** *Let $\alpha$ be a parameter in $(0, 1)$, and let $\mathcal{W} = \{\Phi(\cdot)^T\beta : \beta \in \mathbb{R}^d\}$ denote a class of linear weight functions over a basis $\Phi : \mathcal{X} \times \mathcal{Y} \to \mathbb{R}^d$. Assume that the data $\{(X_i, Y_i)\}_{i\in[n+1]}$ are drawn i.i.d. from a distribution $\mathcal{D}$, $\{\varepsilon_i\}_{i\in[n+1]}$ are drawn i.i.d. from a distribution $\mathcal{D}_{rn}$, independently from the dataset, and the distribution of $\widetilde{S}(X, Y, \varepsilon) \mid \Phi(X, Y)$ is continuous. Then, for any $w \in \mathcal{W}$, the prediction set given by Algorithm 3 satisfies*

$$|\mathbb{E}_{D,E}[\text{wCD}(\mathcal{C}, \alpha, w)]| \leq \frac{d}{n+1}\mathbb{E}_{D_+}\left[\max_{i\in[n+1]}|w(X_i, Y_i)|\right],$$

*where $D$ is the calibration dataset $\{(X_i, Y_i)\}_{i\in[n]}$, $E$ is the corresponding noise $\{\varepsilon_i\}_{i\in[n]}$ and $D_+$ is the full dataset $\{(X_i, Y_i)\}_{i\in[n+1]}$.*

Algorithm 3 is significantly more computationally expensive than Algorithms 1 and 2, as it performs multiple quantile regressions per test point. Let $T_{qr}$ denote the time required for a single quantile regression. In a naive implementation, computing the prediction set scales as $O(|\mathcal{Y}| \cdot T_{qr})$. Even if for certain score and weight functions the construction of the prediction set avoids enumerating $\mathcal{Y}$, the complexity remains $O(T_{qr})$ per test point.

## 5. Experiments

We empirically evaluate the conditional coverage of Kandinsky conformal prediction on real-world tasks with natural groups: income prediction across US states (Ding et al., 2021) and toxic comment detection across demographic groups (Borkan et al., 2019; Koh et al., 2021). The data is divided into a training set for learning the base predictor, a calibration set for learning the conformal predictor, and a test set for evaluation. We repeat all experiments 100 times with reshuffled calibration and test sets.

**Algorithms.** We implement Kandinsky conformal prediction as described in Algorithms 1 and 2 for probabilistic groups. According to Corollary 3.4, we estimate the basis function $\Phi_G$ (see Equation (3)) using a Gradient Boosting Tree classifier. *Kandinsky (XY)* uses the covariates and the label as input to learn the basis ($\phi = (X, Y)$), while *Kandinsky (FY)* uses the output of the predictor and the label ($\phi = (f(X), Y)$). We compare this approach with *split* conformal prediction, *class-conditional* conformal prediction (Löfström et al., 2015) (where labels are discrete), and *conservative* conformal prediction (Barber et al., 2021), which estimates the $(1-\alpha)$-quantile of scores for each group and constructs sets using the score threshold determined by the maximum quantile.

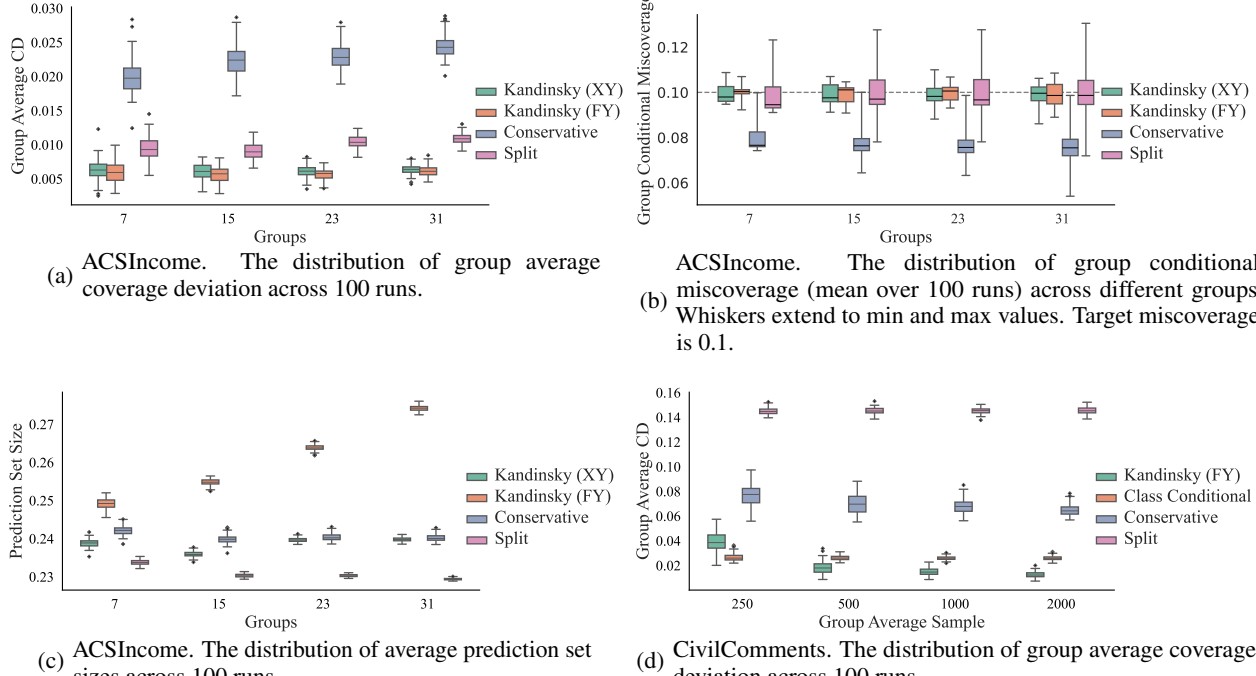

(a) ACSIncome. The distribution of group average coverage deviation across 100 runs.

(b) ACSIncome. The distribution of group conditional miscoverage (mean over 100 runs) across different groups. Whiskers extend to min and max values. Target miscoverage is 0.1.

(c) ACSIncome. The distribution of average prediction set sizes across 100 runs.

(d) CivilComments. The distribution of group average coverage deviation across 100 runs.

Figure 3: Main results. (a)(b)(c) ACSIncome. We vary the number of groups in calibration and test sets while controlling samples per group. (d) CivilComments. We redistribute samples between calibration and test sets to vary the number of calibration samples.

**Metrics.** Given $m$ test samples, we measure the group-conditional miscoverage $\mathrm{M}(G, \mathcal{C})$ for a group $G$ and a prediction set $\mathcal{C}$.

$$\mathrm{M}(G, \mathcal{C}) \coloneqq \frac{\sum_{i=n+1}^{n+m} \mathbb{1}\left\{Y_i \notin \mathcal{C}(X_i, \varepsilon_i)\right\} \cdot \mathbb{1}\left\{Z_i \in G\right\}}{\sum_{i=n+1}^{n+m} \mathbb{1}\left\{Z_i \in G\right\}}.$$

We compute the group average coverage deviation $\mathrm{CD}(\mathcal{C})$.

$$\mathrm{CD}(\mathcal{C}) \coloneqq \frac{1}{|\mathcal{G}|} \sum_{G \in \mathcal{G}} |\mathrm{M}(G, \mathcal{C}) - \alpha|.$$

We also evaluate the prediction set size. For real-valued labels, we divide the label domain into 100 evenly spaced bins. We count the number of bin midpoints that fall within the prediction set and estimate its size by multiplying this number by the bin width.

## 5.1. ACSIncome: A Regression Example

We predict personal income using US Census data collected from different states, aiming to achieve the target coverage conditioned on each state. The state attribute is omitted from the covariates and labels of the test points. We investigate the scalability of the algorithms by varying the number of groups in the calibration and test set, while controlling the number of samples per group. Specifically, we sample

10,000 individuals per state to train a base Gradient Boosting Tree regressor and the basis weight functions for our method. We sample 4,000 individuals per state for calibration and 2,000 individuals for testing. The experiment is performed over 31 states, each with at least 16,000 samples. We use the Conformalized Quantile Regression score function (CQR) (Romano et al., 2019). For further details, see Appendix C.1.

Kandinsky conformal prediction achieves the best group average coverage deviation, as shown in Figure 3(a). Weight functions based on predictors and labels (FY) slightly outperform those based on covariates and labels (XY), as learning a group classifier with lower-dimensional input is easier with finite samples. Both Kandinsky algorithms maintain stable average coverage deviation as the number of groups increases. Figure 3(b) shows that our approach achieves the minimum difference between the maximum and minimum miscoverage across various groups. All algorithms exhibit a larger gap with more groups, aligned with the universal bound on the maximum group coverage deviation (Corollary 3.4), which scales inversely with the relative proportion of subgroups. Kandinsky (XY) produces prediction sets with moderate and stable sizes for various groups numbers, while the size of Kandinsky (FY) scales with the number of groups, as shown in Figure 3(c). This reveals a tradeoff between the computational efficiency of

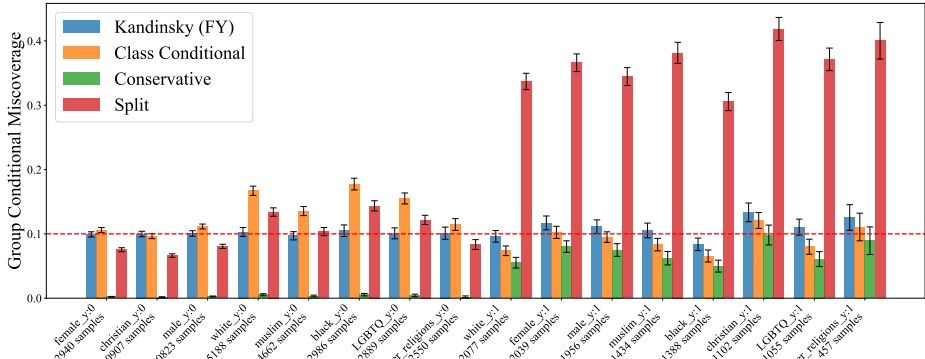

(a) CivilComments. Conditional miscoverage for each group. Target miscoverage is 0.1. Error bars are standard deviations of 100 runs.

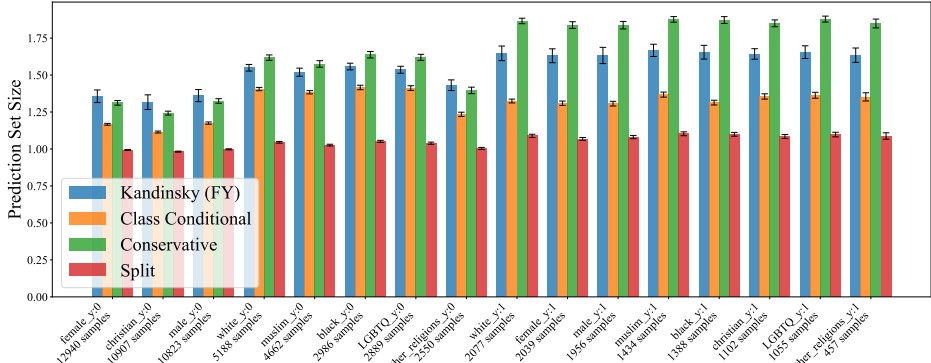

(b) CivilComments. Average prediction set sizes for each group. Error bars are standard deviations of 100 runs.

Figure 4: Group-wise metrics on CivilComments. There are 4000 calibration samples per group on average. We annotate the calibration sample size for each group. Groups with positive labels (toxic) are represented by y:1.

learning a weight function class and the effectiveness of the prediction set.

## 5.2. CivilComments: A Classification Example

The task is to detect whether comments on online articles contain toxic content. Demographic attributes, such as gender, religion, race, and sexual orientation, are sometimes mentioned in the comments. According to the Koh et al. (2021) benchmark, there are 16 overlapping groups jointly defined by eight demographic attributes and two labels. We consider a setup where the conformal prediction algorithm has only indirect access to the training data through a given base predictor, a DistilBERT model finetuned on the training split. Kandinsky conformal prediction (XY) cannot be applied, and the weight functions of Kandinsky (FY) are trained on the calibration set. We use the randomized Adaptive Prediction Sets (APS) score function (Romano et al., 2020b; Ding et al., 2023). We also evaluate the algorithms' adaptability to varying calibration sample sizes by redistributing samples between the calibration and test sets. See Appendix C.2 for more details.

Kandinsky conformal prediction achieves the best group average coverage deviation when the calibration set contains an average of over 500 samples per group, as shown in Figure 3(d). Class-conditional conformal prediction achieves the best performance with 250 samples per group. However, unlike our approach, its coverage deviation does not improve with larger sample sizes. This is because Kandinsky considers the full set of groups, which are more fine-grained than classes. In Figures 4(a) and 4(b), we examine conditional miscoverage and prediction set sizes for each group under the setting of 4000 group average samples. Overall, Kandinsky generates prediction sets with conditional miscoverage closest to the target level for most groups while maintaining moderate sizes. We achieve the best calibrated coverage for groups with more than 2000 samples.

## Acknowledgements

KB's research is supported by NSF awards CCF-2311649, CNS-2232692 and CNS-2247484.

## Impact Statement

In this paper, we introduce Kandinsky Conformal Prediction, which extends conformal prediction to provide coverage guarantees across a broader class of overlapping and fractional groups. This framework addresses fundamental challenges in fairness and distribution shift, ensuring robust, well-calibrated uncertainty quantification across diverse subpopulations. By maintaining computational efficiency, our method strengthens the technical foundation for reliable predictive modeling in real-world applications.

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

# A. Proofs and Remarks from Section 3

In this section, we present the omitted proofs from Section 3 and provide some further discussion about the coverage and the computational complexity of Kandinsky conformal prediction.

## A.1. Proofs from Section 3

We first show in the following Lemma that for a fixed $w \in \mathcal{W}$, the expected weighted coverage deviation is bounded with high probability. Note that this statement provides a bound for a single weight function.

**Lemma A.1.** *Let $\alpha$ and $\delta$ be parameters in $(0, 1)$, and $\mathcal{W} = \{\Phi(\cdot)^T \beta : \beta \in \mathbb{R}^d\}$ denote a class of linear weight functions over a bounded basis $\Phi : \mathcal{X} \times \mathcal{Y} \to \mathbb{R}^d$. Assume that the data $\{(X_i, Y_i)\}_{i \in [n+1]}$ are drawn i.i.d. from a distribution $\mathcal{D}$, $\{\varepsilon_i\}_{i \in [n+1]}$ are drawn i.i.d. from a distribution $\mathcal{D}_{rm}$, independently from the dataset, and the distribution of $\widetilde{S}(X, Y, \varepsilon) \mid \Phi(X, Y)$ is continuous. Then, there exists an absolute constant $C$ such that, for every $w \in \mathcal{W}$, with probability at least $1 - \delta$ over the randomness of the calibration dataset $\{(X_i, Y_i)\}_{i \in [n]}$ and the noise $\{\varepsilon_i\}_{i \in [n]}$, the (randomized) prediction set $\mathcal{C}$, given by Algorithms 1 and 2, satisfies*

$$\left| \mathbb{E}_{(X,Y) \sim \mathcal{D}, \varepsilon \sim \mathcal{D}_m} \left[ w(X, Y) \left( \mathbb{1}\{Y \in \mathcal{C}(X; \varepsilon)\} - (1 - \alpha) \right) \right] \right| \leq \|w\|_\infty \left( C\sqrt{\frac{d}{n}} + \frac{d}{n} + \max\{\alpha, 1 - \alpha\} \sqrt{\frac{2 \ln(4/\delta)}{n}} \right).$$

*Proof.* Denote $\widetilde{S}(X_i, Y_i, \varepsilon_i)$ by $\widetilde{S}_i$. Then $\{(X_i, Y_i, \widetilde{S}_i)\}_{i \in [n]}$ are independent and identically distributed.

Consider the function $L(\eta) = \frac{1}{n} \sum_{i=1}^{n} \ell_\alpha \left( \widehat{q}(X_i, Y_i) + \eta w(X_i, Y_i), \widetilde{S}_i \right)$, $\eta \in \mathbb{R}$, which is convex because $\ell_\alpha(\cdot, \widetilde{S}_i)$ is convex for each $i$.

For any $w \in \mathcal{W}$, since $\widehat{q} + \eta w \in \mathcal{W}$, the subgradients of $L(\eta)$ at $\eta = 0$ satisfy

$$0 \in \partial L(\eta)\big|_{\eta=0} = \left\{ \frac{1}{n} \sum_{i=1}^{n} w(X_i, Y_i)\gamma_i \;\middle|\; \gamma_i = \begin{cases} \alpha - \mathbb{1}\{\widetilde{S}_i > \widehat{q}(X_i, Y_i)\}, & \widehat{q}(X_i, Y_i) \neq \widetilde{S}_i, \\ \alpha_i \in [\alpha - 1, \alpha], & \widehat{q}(X_i, Y_i) = \widetilde{S}_i \end{cases} \right\}.$$

Therefore, there exist $\alpha_i^\star \in [\alpha - 1, \alpha]$, $i \in [n]$ such that

$$\frac{1}{n} \sum_{\widehat{q}(X_i, Y_i) \neq \widetilde{S}_i} w(X_i, Y_i) \left( \alpha - \mathbb{1}\left\{ \widetilde{S}_i > \widehat{q}(X_i, Y_i) \right\} \right) + \frac{1}{n} \sum_{\widehat{q}(X_i, Y_i) = \widetilde{S}_i} \alpha_i^\star w(X_i, Y_i) = 0.$$

This implies

$$\left| \frac{1}{n} \sum_{i=1}^{n} w(X_i, Y_i) \left( \alpha - \mathbb{1}\left\{ \widetilde{S}_i > \widehat{q}(X_i, Y_i) \right\} \right) \right| = \left| \frac{1}{n} \sum_{\widehat{q}(X_i, Y_i) = \widetilde{S}_i} w(X_i, Y_i) \left( \alpha - \alpha_i^\star \right) \right| \tag{4}$$

$$\leq \frac{1}{n} \sum_{i=1}^{n} \mathbb{1}\left\{ \widehat{q}(X_i, Y_i) = \widetilde{S}_i \right\} |w(X_i, Y_i)|. \tag{5}$$

For the left hand side of Equation (4), we consider the function classes

$$\mathcal{H}_1 = \left\{ h_1(x, y, s) = \mathbb{1}\{q(x, y) - s < 0\} \;\middle|\; q \in \mathcal{W} \right\} = \left\{ h_1(x, y, s) = \mathbb{1}\left\{ \Phi(x, y)^T \beta - s < 0 \right\} \;\middle|\; \beta \in \mathbb{R}^d \right\},$$

$$\mathcal{H}_2 = \left\{ h_2(x, y, s) = w(x, y)(\alpha - h_1(x, y, s)) \;\middle|\; h_1 \in \mathcal{H}_1 \right\}.$$

Under the mapping $(x, y, s) \mapsto \begin{bmatrix} \Phi(x, y) \\ -s \end{bmatrix} \in \mathbb{R}^{d+1}$, $\mathcal{H}_1$ is the subset of all homogeneous halfspaces in $\mathbb{R}_{d+1}$, with the normal vector $\begin{bmatrix} \beta \\ 1 \end{bmatrix}$. Therefore, the VC-dimension of $\mathcal{H}_1$ is at most $d + 1$.

For samples $D = \{(X_i, Y_i, \widetilde{S}_i)\}_{i \in [n]}$, the empirical Rademacher complexity of a function class $\mathcal{H}$ restricted to $D$ is defined as

$$\mathcal{R}(\mathcal{H} \circ D) = \frac{1}{n} \mathbb{E}_{\boldsymbol{\sigma}} \left[ \sup_{h \in \mathcal{H}} \sum_{i=1}^{n} \sigma_i h(X_i, Y_i, \widetilde{S}_i) \right],$$

where random variables in $\boldsymbol{\sigma}$ are independent and identically distributed according to $\mathbb{P}[\sigma_i = 1] = \mathbb{P}[\sigma_i = -1] = \frac{1}{2}$.

Wainwright (2019) (Example 5.24) gives an upper bound of the Rademacher complexity for VC classes. There exists an absolute constant $C'$, such that

$$\mathcal{R}(\mathcal{H}_1 \circ D) \leq \frac{1}{n} \mathbb{E}_{\boldsymbol{\sigma}} \left[ \sup_{h \in \mathcal{H}_1} \left| \sum_{i=1}^{n} \sigma_i h(X_i, Y_i, \widetilde{S}_i) \right| \right] \leq C' \sqrt{\frac{d+1}{n}} \leq C' \sqrt{\frac{2d}{n}}.$$

Next, we compute the empirical Rademacher complexity of $\mathcal{H}_2$ restricted to $D$. For any $h, h' \in \mathcal{H}_1$,

$$|w(x,y)(\alpha - h(x,y,s)) - w(x,y)(\alpha - h'(x,y,s))| \leq \|w\|_\infty |h(x,y,s) - h'(x,y,s)|.$$

According to the contraction lemma (Lemma 26.9 in Shalev-Shwartz & Ben-David (2014)),

$$\mathcal{R}(\mathcal{H}_2 \circ D) \leq \|w\|_\infty \mathcal{R}(\mathcal{H}_1 \circ D) \leq C' \|w\|_\infty \sqrt{\frac{2d}{n}}.$$

Therefore, we have a universal generalization bound for the functions in $\mathcal{H}_2$. The functions in $\mathcal{H}_2$ are bounded by $\max\{\alpha, 1-\alpha\} \|w\|_\infty$. According to Theorem 26.5 in Shalev-Shwartz & Ben-David (2014), with probability at least $1 - \delta$ over the randomness in $D = \{(X_i, Y_i, \widetilde{S}_i)\}_{i \in [n]}$, for all $h \in \mathcal{H}_2$,

$$\mathbb{E}\left[h(X, Y, \widetilde{S})\right] - \frac{1}{n} \sum_{i=1}^{n} h(X_i, Y_i, \widetilde{S}_i) \leq 2\mathbb{E}_D\left[\mathcal{R}(\mathcal{H}_2 \circ D)\right] + \max\{\alpha, 1-\alpha\} \|w\|_\infty \sqrt{\frac{2\ln(2/\delta)}{n}}$$

$$\leq 2C' \|w\|_\infty \sqrt{\frac{2d}{n}} + \max\{\alpha, 1-\alpha\} \|w\|_\infty \sqrt{\frac{2\ln(2/\delta)}{n}}.$$

Symmetrically, we can replace $h$ with $-h$ in the class $\mathcal{H}_2$. Since $\mathcal{R}(\mathcal{H}_2 \circ D) = \mathcal{R}(-\mathcal{H}_2 \circ D)$, with probability at least $1 - \delta$, for all $h \in \mathcal{H}_2$,

$$\frac{1}{n} \sum_{i=1}^{n} h(X_i, Y_i, \widetilde{S}_i) - \mathbb{E}\left[h(X, Y, \widetilde{S})\right] \leq 2C' \|w\|_\infty \sqrt{\frac{2d}{n}} + \max\{\alpha, 1-\alpha\} \|w\|_\infty \sqrt{\frac{2\ln(2/\delta)}{n}}.$$

Taking the union bound, with probability at least $1 - \delta$ over the randomness in $D$, for all $h \in \mathcal{H}_2$,

$$\left| \mathbb{E}\left[h(X, Y, \widetilde{S})\right] - \frac{1}{n} \sum_{i=1}^{n} h(X_i, Y_i, \widetilde{S}_i) \right| \leq 2C' \|w\|_\infty \sqrt{\frac{2d}{n}} + \max\{\alpha, 1-\alpha\} \|w\|_\infty \sqrt{\frac{2\ln(4/\delta)}{n}}.$$

By expanding functions in $\mathcal{H}_2$ as $h(x,y,s) = w(x,y)\left(\alpha - \mathbb{1}\{q(x,y) - s < 0\}\right)$ where $q \in \mathcal{W}$, with probability at least $1 - \delta$ over the randomness in $D$, for all $q \in \mathcal{W}$,

$$\left| \mathbb{E}\left[w(X,Y)\left(\mathbb{1}\left\{\widetilde{S} \leq q(X,Y)\right\} - (1-\alpha)\right)\right] - \frac{1}{n} \sum_{i=1}^{n} w(X_i, Y_i)\left(\alpha - \mathbb{1}\left\{q(X_i, Y_i) - \widetilde{S}_i < 0\right\}\right) \right| =$$

$$\left| \mathbb{E}\left[w(X,Y)\left(\alpha - \mathbb{1}\left\{q(X,Y) - \widetilde{S} < 0\right\}\right)\right] - \frac{1}{n} \sum_{i=1}^{n} w(X_i, Y_i)\left(\alpha - \mathbb{1}\left\{q(X_i, Y_i) - \widetilde{S}_i < 0\right\}\right) \right| \leq$$

$$2C' \|w\|_\infty \sqrt{\frac{2d}{n}} + \max\{\alpha, 1-\alpha\} \|w\|_\infty \sqrt{\frac{2\ln(4/\delta)}{n}}.$$

In particular, this inequality holds for $\widehat{q}$. By plugging in Equation (4) and Equation (5),

$$\left| \mathbb{E}_{(X,Y)\sim\mathcal{D}, \varepsilon\sim\mathcal{D}_m} \left[ w(X,Y) \left( \mathbb{1}\left\{ \widetilde{S} \le \widehat{q}(X,Y) \right\} - (1-\alpha) \right) \right] \right| \le$$

$$\left| \frac{1}{n} \sum_{i=1}^{n} w(X_i, Y_i) \left( \alpha - \mathbb{1}\left\{ \widehat{q}(X_i, Y_i) - \widetilde{S}_i < 0 \right\} \right) \right| + 2C'\|w\|_\infty \sqrt{\frac{2d}{n}} + \max\{\alpha, 1-\alpha\} \|w\|_\infty \sqrt{\frac{2\ln(4/\delta)}{n}} \le$$

$$\frac{1}{n} \sum_{i=1}^{n} \mathbb{1}\left\{ \widehat{q}(X_i.Y_i) = \widetilde{S}_i \right\} |w(X_i, Y_i)| + 2C'\|w\|_\infty \sqrt{\frac{2d}{n}} + \max\{\alpha, 1-\alpha\} \|w\|_\infty \sqrt{\frac{2\ln(4/\delta)}{n}} \le \qquad (6)$$

$$\frac{1}{n} \|w\|_\infty \sum_{i=1}^{n} \mathbb{1}\left\{ \widehat{q}(X_i.Y_i) = \widetilde{S}_i \right\} + 2C'\|w\|_\infty \sqrt{\frac{2d}{n}} + \max\{\alpha, 1-\alpha\} \|w\|_\infty \sqrt{\frac{2\ln(4/\delta)}{n}}.$$

Theorem 3 in Gibbs et al. (2023) provides an upper bound for a quantity similar to $\sum_{i=1}^{n} \mathbb{1}\left\{ \widehat{q}(X_i.Y_i) = \widetilde{S}_i \right\}$, under a different assumption that the distribution of $\widetilde{S} \mid X$ is continuous. We will follow a similar proof technique, but we assume that the distribution of $\widetilde{S} \mid \Phi$ is continuous.

There exists $\widehat{\beta} \in \mathbb{R}^d$ such that $\widehat{q} = \Phi(\cdot)^T \widehat{\beta}$.

$$\mathbb{P}\left[ \sum_{i=1}^{n} \mathbb{1}\left\{ \widehat{q}(X_i.Y_i) = \widetilde{S}_i \right\} \ge d+1 \right] =$$

$$\mathbb{E}\left[ \mathbb{P}\left[ \sum_{i=1}^{n} \mathbb{1}\left\{ \Phi(X_i, Y_i)^T \widehat{\beta} = \widetilde{S}_i \right\} \ge d+1 \;\middle|\; \Phi(X_1, Y_1), ..., \Phi(X_n, Y_n) \right] \right] =$$

$$\mathbb{E}\left[ \mathbb{P}\left[ \exists\, 1 \le j_1 \le ... \le j_{d+1} \le n, \forall\, 1 \le k \le d+1, \Phi(X_{j_k}, Y_{j_k})^T \widehat{\beta} = \widetilde{S}_{j_k} \;\middle|\; \Phi(X_1, Y_1), ..., \Phi(X_n, Y_n) \right] \right] \le$$

$$\mathbb{E}\left[ \sum_{1 \le j_1 < ... < j_{d+1} \le n} \mathbb{P}\left[ \exists\, \beta \in \mathbb{R}^d, \forall\, 1 \le k \le d+1, \Phi(X_{j_k}, Y_{j_k})^T \beta = \widetilde{S}_{j_k} \;\middle|\; \Phi(X_1, Y_1), ..., \Phi(X_n, Y_n) \right] \right] \le$$

$$\mathbb{E}\left[ \sum_{1 \le j_1 < ... < j_{d+1} \le n} \mathbb{P}\left[ \left( \widetilde{S}_{j_1}, ..., \widetilde{S}_{j_{d+1}} \right) \in \text{RowSpace}\left( \left[ \Phi(X_{j_1}, Y_{j_1}), ..., \Phi(X_{j_{d+1}}, Y_{j_{d+1}}) \right] \right) \right. \right.$$

$$\left. \left. \phantom{\sum} \;\middle|\; \Phi(X_1, Y_1), ..., \Phi(X_n, Y_n) \right] \right] =$$

$$0.$$

The last equation holds because the distribution of $\left( \widetilde{S}_{j_1}, ..., \widetilde{S}_{j_{d+1}} \right) \;\middle|\; \Phi(X_1, Y_1), ..., \Phi(X_n, Y_n)$ is continuous, and RowSpace $\left( \left[ \Phi(X_{j_1}, Y_{j_1}), ..., \Phi(X_{j_{d+1}}, Y_{j_{d+1}}) \right] \right)$ is at most $d-$dimensional while $\left( \widetilde{S}_{j_1}, ..., \widetilde{S}_{j_{d+1}} \right)$ is a $(d+1)-$dimensional vector. Therefore, with probability 1,

$$\sum_{i=1}^{n} \mathbb{1}\left\{ \widehat{q}(X_i.Y_i) = \widetilde{S}_i \right\} \le d.$$

Plugging the inequality into Equation (6), with probability at least $1 - \delta$ over the randomness in $D$,

$$\left| \mathbb{E}_{(X,Y)\sim\mathcal{D}, \varepsilon\sim\mathcal{D}_m} \left[ w(X,Y) \left( \mathbb{1}\left\{ \widetilde{S} \le \widehat{q}(X,Y) \right\} - (1-\alpha) \right) \right] \right| \le$$

$$\|w\|_\infty \left( \frac{d}{n} + 2C' \sqrt{\frac{2d}{n}} + \max\{\alpha, 1-\alpha\} \sqrt{\frac{2\ln(4/\delta)}{n}} \right).$$

$$\square$$

Using Lemma A.1 and a union bound, we prove a high probability bound on the expected coverage deviation for all weight functions in $\mathcal{W}$.

**Theorem 3.1** (Restated). Let $\alpha$ and $\delta$ be parameters in $(0,1)$, and $\mathcal{W} = \{\Phi(\cdot)^T\beta : \beta \in \mathbb{R}^d\}$ denote a class of linear weight functions over a bounded basis $\Phi : \mathcal{X} \times \mathcal{Y} \to \mathbb{R}^d$. Without loss of generality, assume $\|\Phi(x,y)\|_\infty \leq 1$ for all $x, y$. Assume that the data $\{(X_i, Y_i)\}_{i \in [n]}$ are drawn i.i.d. from a distribution $\mathcal{D}$, $\{\varepsilon_i\}_{i \in [n]}$ are drawn i.i.d. from a distribution $\mathcal{D}_{\mathrm{rn}}$, independently from the dataset, and the distribution of $\widetilde{S}(X, Y, \varepsilon) \mid \Phi(X, Y)$ is continuous. Then, there exists an absolute constant $C$ such that, with probability at least $1 - \delta$ over the randomness of the calibration dataset $\{(X_i, Y_i)\}_{i \in [n]}$ and the noise $\{\varepsilon_i\}_{i \in [n]}$, the (randomized) prediction set $\mathcal{C}$, given by Algorithms 1 and 2, satisfies

$$\left| \mathbb{E}_{(X,Y) \sim \mathcal{D}, \varepsilon \sim \mathcal{D}_{\mathrm{rn}}} \left[ w_\beta(X, Y) \left( \mathbb{1}\{Y \in \mathcal{C}(X; \varepsilon)\} - (1 - \alpha) \right) \right] \right| \leq \|\beta\|_1 \left( C\sqrt{\frac{d}{n}} + \frac{d}{n} + \max\{\alpha, 1 - \alpha\}\sqrt{\frac{2\ln(4d/\delta)}{n}} \right),$$

for every $w_\beta = \Phi(\cdot)^T\beta$.

*Proof.* Let $\Phi(\cdot) = (\phi_1(\cdot), \cdots, \phi_d(\cdot))^T$, where $\phi_j \in \mathbb{R}^{\mathcal{X} \times \mathcal{Y}}$ for $1 \leq j \leq d$. According to Lemma A.1, for all $1 \leq j \leq d$, with probability at least $1 - \delta/d$,

$$\left| \mathbb{E}_{(X,Y) \sim \mathcal{D}, \varepsilon \sim \mathcal{D}_{\mathrm{rn}}} \left[ \phi_j(X, Y) \left( \mathbb{1}\{Y \in \mathcal{C}(X; \varepsilon)\} - (1 - \alpha) \right) \right] \right| \leq C\sqrt{\frac{d}{n}} + \frac{d}{n} + \max\{\alpha, 1 - \alpha\}\sqrt{\frac{2\ln(4d/\delta)}{n}}.$$

Then, we have the union bound. With probability at least $1 - \delta$, for all $1 \leq j \leq d$,

$$\left| \mathbb{E}_{(X,Y) \sim \mathcal{D}, \varepsilon \sim \mathcal{D}_{\mathrm{rn}}} \left[ \phi_j(X, Y) \left( \mathbb{1}\{Y \in \mathcal{C}(X; \varepsilon)\} - (1 - \alpha) \right) \right] \right| \leq C\sqrt{\frac{d}{n}} + \frac{d}{n} + \max\{\alpha, 1 - \alpha\}\sqrt{\frac{2\ln(4d/\delta)}{n}}.$$

For any $w_\beta = \Phi(\cdot)^T\beta$,

$$\left| \mathbb{E}_{(X,Y) \sim \mathcal{D}, \varepsilon \sim \mathcal{D}_{\mathrm{rn}}} \left[ w_\beta(X, Y) \left( \mathbb{1}\{Y \in \mathcal{C}(X; \varepsilon)\} - (1 - \alpha) \right) \right] \right| \leq$$
$$\sum_{j=1}^d \left| \mathbb{E}_{(X,Y) \sim \mathcal{D}, \varepsilon \sim \mathcal{D}_{\mathrm{rn}}} \left[ \beta_j \phi_j(X, Y) \left( \mathbb{1}\{Y \in \mathcal{C}(X; \varepsilon)\} - (1 - \alpha) \right) \right] \right| \leq$$
$$\sum_{j=1}^d |\beta_j| \cdot \left( C\sqrt{\frac{d}{n}} + \frac{d}{n} + \max\{\alpha, 1 - \alpha\}\sqrt{\frac{2\ln(4d/\delta)}{n}} \right) =$$
$$\|\beta\|_1 \left( C\sqrt{\frac{d}{n}} + \frac{d}{n} + \max\{\alpha, 1 - \alpha\}\sqrt{\frac{2\ln(4d/\delta)}{n}} \right).$$

$\square$

**Corollary 3.2** (Restated). Let $\alpha, \delta, \mathcal{W}, w_\beta$ be specified as in Theorem 3.1. Assume that $\{(X_i, Y_i)\}_{i \in [n+1]}$ are drawn i.i.d. from a distribution $\mathcal{D}$, $\{\varepsilon_i\}_{i \in [n+1]}$ are drawn i.i.d. from a distribution $\mathcal{D}_{\mathrm{rn}}$, independently from the dataset, and the distribution of $\widetilde{S}(X, Y, \varepsilon) \mid \Phi(X, Y)$ is continuous. Then, there exists an absolute constant $C$ such that the (randomized) prediction set $\mathcal{C}$, given by Algorithms 1 and 2, satisfies

$$\mathbb{E}_{D,E} \left[ \sup_{w_\beta \in \mathcal{W}} \mathbb{E}_{(X,Y) \sim \mathcal{D}, \varepsilon \sim \mathcal{D}_{\mathrm{rn}}} \left[ \frac{w_\beta(X, Y)}{\|\beta\|_1} \left( \mathbb{1}\{Y \in \mathcal{C}(X; \varepsilon)\} - (1 - \alpha) \right) \right] \right] \leq C\sqrt{\frac{d}{n}} + \frac{d}{n},$$

where $D$ is the calibration dataset $\{(X_i, Y_i)\}_{i \in [n]}$ and $E$ is the corresponding noise $\{\varepsilon_i\}_{i \in [n]}$.

*Proof.* According to Theorem 3.1, there exists an absolute constant $C'$ such that, with probability at least $1 - \delta$ over the

randomness in $D$ and $E$,

$$\sup_{w_\beta \in \mathcal{W}} \mathbb{E}_{(X,Y) \sim \mathcal{D}, \varepsilon \sim \mathcal{D}_{\mathrm{m}}} \left[ \frac{w_\beta(X,Y)}{\|\beta\|_1} \left( \mathbb{1}\{Y \in \mathcal{C}(X;\varepsilon)\} - (1-\alpha) \right) \right] \le$$

$$\sup_{w_\beta \in \mathcal{W}} \left| \mathbb{E}_{(X,Y) \sim \mathcal{D}, \varepsilon \sim \mathcal{D}_{\mathrm{m}}} \left[ \frac{w_\beta(X,Y)}{\|\beta\|_1} \left( \mathbb{1}\{Y \in \mathcal{C}(X;\varepsilon)\} - (1-\alpha) \right) \right] \right| \le$$

$$C'\sqrt{\frac{d}{n}} + \frac{d}{n} + \max\{\alpha, 1-\alpha\}\sqrt{\frac{2\ln(4d/\delta)}{n}} \le$$

$$C'\sqrt{\frac{d}{n}} + \frac{d}{n} + \max\{\alpha, 1-\alpha\}\sqrt{\frac{2\ln d}{n}} + \max\{\alpha, 1-\alpha\}\sqrt{\frac{2\ln(4/\delta)}{n}}.$$

Let the random variable $M_{D,E} = \sup_{w_\beta \in \mathcal{W}} \mathbb{E}_{(X,Y) \sim \mathcal{D}, \varepsilon \sim \mathcal{D}_{\mathrm{m}}} \left[ \frac{w_\beta(X,Y)}{\|\beta\|_1} \left( \mathbb{1}\{Y \in \mathcal{C}(X;\varepsilon)\} - (1-\alpha) \right) \right]$. We have

$$\mathbb{P}_{D,E}\left[ M_{D,E} > C'\sqrt{\frac{d}{n}} + \frac{d}{n} + \max\{\alpha, 1-\alpha\}\sqrt{\frac{2\ln d}{n}} + \max\{\alpha, 1-\alpha\}\sqrt{\frac{2\ln(4/\delta)}{n}} \right] \le \delta.$$

By inverting the bound, for $t > C'\sqrt{\frac{d}{n}} + \frac{d}{n} + \max\{\alpha, 1-\alpha\}\sqrt{\frac{2\ln d}{n}}$,

$$\mathbb{P}_{D,E}[M_{D,E} > t] \le 4\exp\left[ -\frac{n\left(t - C'\sqrt{\frac{d}{n}} - \frac{d}{n} - \max\{\alpha, 1-\alpha\}\sqrt{\frac{2\ln d}{n}}\right)^2}{2\max\{\alpha, 1-\alpha\}^2} \right].$$

Let $I = C'\sqrt{\frac{d}{n}} + \frac{d}{n} + \max\{\alpha, 1-\alpha\}\sqrt{\frac{2\ln d}{n}}$. Since $M_{D,E} \ge 0$,

$$\begin{aligned}
\mathbb{E}_{D,E}[M_{D,E}] &= \int_{t=0}^{+\infty} \mathbb{P}_{D,E}[M_{D,E} > t]\, \mathrm{d}t \\
&= \int_{t=0}^{I} \mathbb{P}_{D,E}[M_{D,E} > t]\, \mathrm{d}t + \int_{t=I}^{+\infty} \mathbb{P}_{D,E}[M_{D,E} > t]\, \mathrm{d}t \\
&\le C'\sqrt{\frac{d}{n}} + \frac{d}{n} + \max\{\alpha, 1-\alpha\}\sqrt{\frac{2\ln d}{n}} + \int_{t=I}^{+\infty} \mathbb{P}_{D,E}[M_{D,E} > t]\, \mathrm{d}t \\
&\le C'\sqrt{\frac{d}{n}} + \frac{d}{n} + \max\{\alpha, 1-\alpha\}\sqrt{\frac{2\ln d}{n}} + \int_{t=I}^{+\infty} 4\exp\left[ -\frac{n(t-I)^2}{2\max\{\alpha, 1-\alpha\}^2} \right] \mathrm{d}t \\
&= C'\sqrt{\frac{d}{n}} + \frac{d}{n} + \max\{\alpha, 1-\alpha\}\sqrt{\frac{2\ln d}{n}} + \max\{\alpha, 1-\alpha\}\sqrt{\frac{8\pi}{n}} \\
&\le \left( C' + \sqrt{2} + \sqrt{8\pi} \right)\sqrt{\frac{d}{n}} + \frac{d}{n}.
\end{aligned}$$

$\square$

**Corollary 3.4** (Restated). *Let $\alpha, \delta \in (0,1)$, $\phi(X,Y)$ be a sufficient statistic for $\widetilde{S}$, such that $\widetilde{S}$ is conditionally independent of $X, Y$ given $\phi$, and $\mathcal{W} = \{\sum_{G \in \mathcal{G}} \beta_G \Phi_G : \beta_G \in \mathbb{R}, \forall\, G \in \mathcal{G}\}$. Assume that the data $\{(X_i, Y_i)\}_{i \in [n+1]}$ are drawn i.i.d. from a distribution $\mathcal{D}$, $\{\varepsilon_i\}_{i \in [n+1]}$ are drawn i.i.d. from a distribution $\mathcal{D}_{\mathrm{m}}$, independently from the dataset, and the distribution of $\widetilde{S}(X, Y, \varepsilon) \mid \Phi(X, Y)$ is continuous. There exists an absolute constant $C$ such that, with probability at least $1 - \delta$ over the randomness of the calibration dataset $\{(X_i, Y_i)\}_{i \in [n]}$ and the noise $\{\varepsilon_i\}_{i \in [n]}$, the (randomized) prediction set $\mathcal{C}$ given by Algorithms 1 and 2 satisfies, for all $G \in \mathcal{G}$,*

$$\left| \mathbb{P}\left[ Y_{n+1} \in \mathcal{C}(X_{n+1}; \varepsilon_{n+1}) \mid Z_{n+1} \in G, \{X_i, Y_i, \varepsilon_i\}_{i \in [n]} \right] - (1-\alpha) \right| \le$$

$$\frac{1}{\mathbb{P}[Z_{n+1} \in G]} \left( C\sqrt{\frac{|\mathcal{G}|}{n}} + \frac{|\mathcal{G}|}{n} + \max\{\alpha, 1-\alpha\}\sqrt{\frac{2\ln(4|\mathcal{G}|/\delta)}{n}} \right).$$

*Proof.* Since $\widetilde{S}$ is conditionally independent of $X, Y$ given $\phi$, more formally $\widetilde{S} \perp\!\!\!\perp X, Y \mid \phi(X, Y)$, and $\varepsilon \perp\!\!\!\perp X, Y$, we have $\widetilde{S}, \varepsilon \perp\!\!\!\perp X, Y \mid \phi(X, Y)$. Therefore $\widetilde{S} \perp\!\!\!\perp X, Y \mid \phi(X, Y), \varepsilon$. This implies for any $x_1, y_1, x_2, y_2$ such that $\phi(x_1, y_1) = \phi(x_2, y_2)$, we have $\widetilde{S}(x_1, y_1, \varepsilon) = \widetilde{S}(x_2, y_2, \varepsilon)$. Therefore, $\widetilde{S}(X, Y, \varepsilon)$ is measurable w.r.t. $\phi(X, Y), \varepsilon$. From the definition of $\mathcal{W}$, any $w(X, Y)$ with $w \in \mathcal{W}$ is also measurable w.r.t. $\phi(X, Y), \varepsilon$.

According to Theorem 3.1, there exists an absolute constant $C$ such that, with probability at least $1 - \delta$ over the randomness in $\{(X_i, Y_i)\}_{i \in [n]}$ and $\{\varepsilon_i\}_{i \in [n]}$, for every $w_\beta \in \mathcal{W}$,

$$\left| \mathbb{E}_{X,Y,\varepsilon} \left[ w_\beta(X, Y) \left( \mathbb{1}\left\{ \widetilde{S}(X, Y, \varepsilon) \leq \widehat{q}(X, Y) \right\} - (1 - \alpha) \right) \right] \right| =$$
$$\left| \mathbb{E}_{X,Y,\varepsilon} \left[ w_\beta(X, Y) \left( \mathbb{1}\left\{ Y \in \mathcal{C}(X; \varepsilon) \right\} - (1 - \alpha) \right) \right] \right| \leq$$
$$\|\beta\|_1 \left( C\sqrt{\frac{d}{n}} + \frac{d}{n} + \max\{\alpha, 1 - \alpha\} \sqrt{\frac{2 \ln(4d/\delta)}{n}} \right) =$$
$$\|\beta\|_1 \left( C\sqrt{\frac{|\mathcal{G}|}{n}} + \frac{|\mathcal{G}|}{n} + \max\{\alpha, 1 - \alpha\} \sqrt{\frac{2 \ln(4|\mathcal{G}|/\delta)}{n}} \right).$$

For any $G \in \mathcal{G}$, by taking $w_\beta(x, y) = \frac{1}{\mathbb{P}[Z \in G]} \mathbb{P}[Z \in G \mid \phi(X, Y) = \phi(x, y)]$, we have $\|\beta\|_1 = \frac{1}{\mathbb{P}[Z_{n+1} \in G]}$. Since $\widetilde{S}(X, Y, \varepsilon), \widehat{q}(X, Y)$ are measurable w.r.t. $\phi(X, Y), \varepsilon$, by Bayes formula,

$$\mathbb{E}_{X,Y,\varepsilon} \left[ w_\beta(X, Y) \left( \mathbb{1}\left\{ \widetilde{S}(X, Y, \varepsilon) \leq \widehat{q}(X, Y) \right\} - (1 - \alpha) \right) \right] =$$
$$\mathbb{E}_{X,Y,\varepsilon} \left[ \frac{\mathbb{P}[Z \in G \mid \phi(X, Y)]}{\mathbb{P}[Z \in G]} \left( \mathbb{1}\left\{ \widetilde{S}(X, Y, \varepsilon) \leq \widehat{q}(X, Y) \right\} - (1 - \alpha) \right) \right] =$$
$$\mathbb{E}_{\phi(X,Y),\varepsilon} \left[ \frac{\mathbb{P}[Z \in G \mid \phi(X, Y), \varepsilon]}{\mathbb{P}[Z \in G \mid \varepsilon]} \cdot \left( \mathbb{1}\left\{ \widetilde{S}(X, Y, \varepsilon) \leq \widehat{q}(X, Y) \right\} - (1 - \alpha) \right) \right] =$$
$$\mathbb{E}_{\phi(X,Y),\varepsilon} \left[ \mathbb{1}\left\{ \widetilde{S}(X, Y, \varepsilon) \leq \widehat{q}(X, Y) \right\} - (1 - \alpha) \mid Z \in G \right] =$$
$$\mathbb{E} \left[ \mathbb{1}\left\{ \widetilde{S}(X_{n+1}, Y_{n+1}, \varepsilon_{n+1}) \leq \widehat{q}(X_{n+1}, Y_{n+1}) \right\} - (1 - \alpha) \mid Z_{n+1} \in G, \{X_i, Y_i, \varepsilon_i\}_{i \in [n]} \right] =$$
$$\mathbb{P} \left[ Y_{n+1} \in \mathcal{C}(X_{n+1}; \varepsilon_{n+1}) \mid Z_{n+1} \in G, \{X_i, Y_i, \varepsilon_i\}_{i \in [n]} \right] - (1 - \alpha).$$

Combining the two equations above, the proof is complete. $\qquad\square$

**Corollary 3.3** (Restated). Let parameters $\alpha, \delta \in (0, 1)$, and $\mathcal{W} = \{\sum_{G \in \mathcal{G}} \beta_G \mathbb{1}\{(x, y) \in G\} : \beta_G \in \mathbb{R}, \forall G \in \mathcal{G}\}$. Assume that the data $\{(X_i, Y_i)\}_{i \in [n+1]}$ are drawn i.i.d. from a distribution $\mathcal{D}$, $\{\varepsilon_i\}_{i \in [n+1]}$ are drawn i.i.d. from a distribution $\mathcal{D}_{\mathrm{rn}}$, independently from the dataset, and the distribution of $\widetilde{S}(X, Y, \varepsilon) \mid \Phi(X, Y)$ is continuous. There exists an absolute constant $C$ such that, with probability at least $1 - \delta$ over the randomness of the calibration dataset $\{(X_i, Y_i)\}_{i \in [n]}$ and the noise $\{\varepsilon_i\}_{i \in [n]}$, the (randomized) prediction set $\mathcal{C}$, given by Algorithms 1 and 2 satisfies, for all $G \in \mathcal{G}$,

$$\left| \mathbb{P}[Y_{n+1} \in \mathcal{C}(X_{n+1}; \varepsilon_{n+1}) \mid (X_{n+1}, Y_{n+1}) \in G, \{X_i, Y_i, \varepsilon_i\}_{i \in [n]}] - (1 - \alpha) \right| \leq$$
$$\frac{1}{\mathbb{P}[(X_{n+1}, Y_{n+1}) \in G]} \left( C\sqrt{\frac{|\mathcal{G}|}{n}} + \frac{|\mathcal{G}|}{n} + \max\{\alpha, 1 - \alpha\} \sqrt{\frac{2 \ln(4|\mathcal{G}|/\delta)}{n}} \right).$$

*Proof.* Corollary 3.3 follows directly from Corollary 3.4. $\qquad\square$

**Corollary 3.5** (Restated). Let $\alpha$ and $\delta$ be parameters in $(0, 1)$, and $\mathcal{W} = \{\Phi(\cdot)^T \beta : \beta \in \mathbb{R}^d\}$ denote a class of linear weight functions over a bounded basis $\Phi : \mathcal{X} \times \mathcal{Y} \to \mathbb{R}^d$. Assume that the data $\{(X_i, Y_i)\}_{i \in [n]}$ are drawn i.i.d. from a distribution $\mathcal{D}$, $\{\varepsilon_i\}_{i \in [n+1]}$ are drawn i.i.d. from a distribution $\mathcal{D}_{\mathrm{rn}}$, independently from the dataset, and the distribution of $\widetilde{S}(X, Y, \varepsilon) \mid \Phi(X, Y)$ is continuous. Then, there exists an absolute constant $C$ such that, for every distribution $\mathcal{D}_{\mathrm{T}}$ such that $\frac{d\mathbb{P}_{\mathcal{D}_{\mathrm{T}}}}{d\mathbb{P}_{\mathcal{D}}} \in \mathcal{W}$ and $\left| \frac{d\mathbb{P}_{\mathcal{D}_{\mathrm{T}}}}{d\mathbb{P}_{\mathcal{D}}}(x, y) \right| \leq B$ for any $x \in \mathcal{X}$ and $y \in \mathcal{Y}$, with probability at least $1 - \delta$ over the randomness of the calibration dataset $\{(X_i, Y_i)\}_{i \in [n]}$ and the noise $\{\varepsilon_i\}_{i \in [n]}$, the (randomized) prediction set $\mathcal{C}$, given by Algorithms 1 and 2,

satisfies

$$\left|\mathbb{P}[Y_{n+1} \in \mathcal{C}(X_{n+1}; \varepsilon_{n+1}) \mid \{X_i, Y_i, \varepsilon_i\}_{i \in [n]}] - (1 - \alpha)\right| \leq B\left(C\sqrt{\frac{d}{n}} + \frac{d}{n} + \max\{\alpha, 1 - \alpha\}\sqrt{\frac{2\ln(4/\delta)}{n}}\right),$$

where $(X_{n+1}, Y_{n+1})$ are drawn independently from the distribution $\mathcal{D}_{\mathrm{T}}$.

*Proof.* This corollary follows from Lemma A.1 for any weight function $w \in \mathcal{W}$ of the form $w(x, y) = \frac{d\mathbb{P}_{\mathcal{D}_T}}{d\mathbb{P}_{\mathcal{D}}}(x, y)$ with $\|w\|_\infty \leq B$. $\square$

### A.2. Comparison with Previous Results

For weight functions defined only on the covariates, Kandinsky conformal prediction obtains the same type of guarantees studied in Jung et al. (2023); Gibbs et al. (2023); Areces et al. (2024). Jung et al. (2023) and Areces et al. (2024) essentially implement the same quantile regression method as that described in Algorithms 1 and 2 for the corresponding class of weight functions and for deterministic scores. Typically, for covariate-based weight functions the distribution of $S(X, Y) \mid \Phi(X)$ is continuous and, hence, it is common to set $\widetilde{S}(x, y, \varepsilon) = S(x, y)$. Jung et al. (2023) focus on the group-conditional case, where the groups are subsets of $\mathcal{X}$, and their analysis of high-probability coverage is less optimal compared to the result in Corollary 3.3. Additionally, our analysis of the expected weighted coverage deviation in Corollary 3.2 provides a tighter upper bound compared to (Areces et al., 2024). Lastly, Gibbs et al. (2023) implement a different quantile regression method that we discuss further in Section 4.

### A.3. Computational Details

In Algorithm 2, the prediction set $\mathcal{C}$ is defined as the subset of all labels in $\mathcal{Y}$ where the score is below the value determined by the quantile function $\widehat{q}$, that varies with $y$. This makes calculating the prediction set from the quantile function $\widehat{q}$ complex for large finite sets of labels, and even more so for continuous label domains. This is a problem that can also be encountered in full conformal prediction. For several applications described in this section and for certain chosen score functions, there exist oracles that, given the quantile function $\widehat{q}$ and the test point $\mathcal{X}_{n+1}$, return the prediction set $\mathcal{C}$.

## B. Proofs from Section 4

In this section, we provide the proof of Theorem 4.1.

**Theorem 4.1** (Restated). Let $\alpha$ be a parameter in $(0, 1)$, and let $\mathcal{W} = \{\Phi(\cdot)^T\beta : \beta \in \mathbb{R}^d\}$ denote a class of linear weight functions over a basis $\Phi : \mathcal{X} \times \mathcal{Y} \to \mathbb{R}^d$. Assume that the data $\{(X_i, Y_i)\}_{i \in [n+1]}$ are drawn i.i.d. from a distribution $\mathcal{D}$, $\{\varepsilon_i\}_{i \in [n+1]}$ are drawn i.i.d. from a distribution $\mathcal{D}_{\mathrm{rn}}$, independently from the dataset, and the distribution of $\widetilde{S}(X, Y, \varepsilon) \mid \Phi(X, Y)$ is continuous. Then, for any $w \in \mathcal{W}$, the prediction set given by Algorithm 3 satisfies

$$|\mathbb{E}_{D,E}[\mathrm{wCD}(\mathcal{C}, \alpha, w)]| \leq \frac{d}{n+1}\mathbb{E}_{D_+}\left[\max_{i \in [n+1]}|w(X_i, Y_i)|\right],$$

where $D$ is the calibration dataset $\{(X_i, Y_i)\}_{i \in [n]}$, $E$ is the corresponding noise $\{\varepsilon_i\}_{i \in [n]}$ and $D_+$ is the full dataset $\{(X_i, Y_i)\}_{i \in [n+1]}$.

*Proof.* This proof follows the techniques developed in (Gibbs et al., 2023). For simplicity, $\widetilde{S}(X_i, Y_i, \varepsilon_i)$ is denoted by $\widetilde{S}_i$. Let $\widehat{q}_{Y_{n+1}}$ be the quantile function Algorithm 3 computes for the true label $Y_{n+1}$. For a fixed $w \in \mathcal{W}$ our objective can be reformulated as

$$\mathbb{E}_{D,E}[\mathrm{wCD}(\mathcal{C}, \alpha, w)] = \mathbb{E}\left[w(X_{n+1}, Y_{n+1})\left(\mathbb{1}\{Y_{n+1} \in \mathcal{C}(X_{n+1}; \varepsilon_{n+1})\} - (1 - \alpha)\right)\right]$$
$$= \mathbb{E}\left[w(X_{n+1}, Y_{n+1})\left(\alpha - \mathbb{1}\left\{\widetilde{S}_{n+1} > \widehat{q}_{Y_{n+1}}(X_{n+1}, Y_{n+1})\right\}\right)\right],$$

where the expectations on the right hand side are taken over the randomness of $\{(X_i, Y_i)\}_{i \in [n+1]}$ and $\{\varepsilon_i\}_{i \in [n+1]}$.

Since the data $\{(X_i, Y_i)\}_{i\in[n+1]}$ are i.i.d., the random noise components $\{\varepsilon_i\}_{i\in[n+1]}$ are also i.i.d., and based on Algorithm 3, $\widehat{q}_{Y_{n+1}}$ is invariant to permutations of $\{(X_i, Y_i)\}_{i\in[n+1]}$, the triples in

$$\{(w(X_i, Y_i), \widehat{q}_{Y_{n+1}}(X_i, Y_i), \widetilde{S}_i, \varepsilon_i))\}_{i\in[n+1]}$$

are exchangeable. Hence, we have that

$$\mathbb{E}_{D,E}[\mathrm{wCD}(\mathcal{C}, \alpha, w)] = \mathbb{E}\left[w(X_{n+1}, Y_{n+1})\left(\alpha - \mathbb{1}\left\{\widetilde{S}_{n+1} > \widehat{q}_{Y_{n+1}}(X_{n+1}, Y_{n+1})\right\}\right)\right]$$

$$= \mathbb{E}\left[\frac{1}{n+1}\sum_{i\in[n+1]} w(X_i, Y_i)\left(\alpha - \mathbb{1}\left\{\widetilde{S}_i > \widehat{q}_{Y_{n+1}}(X_i, Y_i)\right\}\right)\right].$$

Since $\widehat{q}_{Y_{n+1}}$ is a minimizer of the convex optimization problem defined in Algorithm 3, we have that for a fixed $w \in \mathcal{W}$, fixed datapoints $\{(X_i, Y_i)\}_{i\in[n+1]}$, and fixed noise $\{\varepsilon_i\}_{i\in[n+1]}$

$$0 \in \partial_\eta \left(\frac{1}{n+1}\sum_{i\in[n+1]} \ell_\alpha(\widehat{q}_{Y_{n+1}}(X_i, Y_i) + \eta w(X_i, Y_i), \widetilde{S}_i)\right)\Bigg|_{\eta=0}.$$

Computing this subrgradient, we obtain that

$$\partial_\eta \left(\frac{1}{n+1}\sum_{i\in[n+1]} \ell_\alpha(\widehat{q}_{Y_{n+1}}(X_i, Y_i) + \eta w(X_i, Y_i), \widetilde{S}_i)\right) =$$

$$\left\{\frac{1}{n+1}\left(\sum_{i\in[n+1]} w(X_i, Y_i)\left(\alpha - \mathbb{1}\left\{\widetilde{S}_i > \widehat{q}_{Y_{n+1}}(X_i, Y_i)\right\}\right)\mathbb{1}\{\widetilde{S}_i \neq \widehat{q}_{Y_{n+1}}(X_i, Y_i)\} + \right.\right.$$

$$\left.\left.\sum_{i\in[n+1]} v_i w(X_i, Y_i)\mathbb{1}\{\widetilde{S}_i = \widehat{q}_{Y_{n+1}}(X_i, Y_i)\}\right)\Bigg| v_i \in [\alpha - 1, \alpha]\right\}.$$

Let $v_i^*$ be one of the values in $[\alpha - 1, \alpha]$ that set the subgradient to zero. Then, we have that

$$\frac{1}{n+1}\sum_{i\in[n+1]} w(X_i, Y_i)\left(\alpha - \mathbb{1}\{\widetilde{S}_i > \widehat{q}_{Y_{n+1}}(X_i, Y_i)\}\right) =$$

$$\frac{1}{n+1}\sum_{i\in[n]} (\alpha - v_i^*)w(X_i, Y_i)\mathbb{1}\{\widetilde{S}_i = \widehat{q}_{Y_{n+1}}(X_i, Y_i)\}.$$

Going back to our previous computation where we have only fixed $w \in \mathcal{W}$, we apply the equality above to obtain that

$$\mathbb{E}_{D,E}[\mathrm{wCD}(\mathcal{C}, \alpha, w)] = \mathbb{E}\left[\frac{1}{n+1}\sum_{i\in[n+1]} w(X_i, Y_i)\left(\alpha - \mathbb{1}\left\{\widetilde{S}_i > \widehat{q}_{Y_{n+1}}(X_i, Y_i)\right\}\right)\right]$$

$$= \mathbb{E}\left[\frac{1}{n+1}\sum_{i\in[n+1]} (\alpha - v_i^*)w(X_i, Y_i)\mathbb{1}\{\widetilde{S}_i = \widehat{q}_{Y_{n+1}}(X_i, Y_i)\}\right].$$

Now, we want to provide an upper bound for our objective. Recall that from the theorem formulation $D_+ =$

$\{(X_i, Y_i)\}_{i \in [n+1]}$ and let $E_+ = \{\varepsilon_i\}_{i \in [n+1]}$. Since $v_i^* \geq \alpha - 1$, we get that

$$
\begin{aligned}
|\mathbb{E}_{D,E}[\text{wCD}(\mathcal{C}, \alpha, w)]| &= \left| \mathbb{E}\left[ \frac{1}{n+1} \sum_{i \in [n+1]} (\alpha - v_i^*) w(X_i, Y_i) \mathbb{1}\{\widetilde{S}_i = \widehat{q}_{Y_{n+1}}(X_i, Y_i)\} \right] \right| \\
&\leq \mathbb{E}\left[ \frac{1}{n+1} \sum_{i \in [n+1]} |w(X_i, Y_i)| \mathbb{1}\{\widetilde{S}_i = \widehat{q}_{Y_{n+1}}(X_i, Y_i)\} \right] \\
&\leq \mathbb{E}\left[ \left( \max_{j \in [n+1]} |w(X_j, Y_j)| \right) \frac{1}{n+1} \sum_{i \in [n+1]} \mathbb{1}\{\widetilde{S}_i = \widehat{q}_{Y_{n+1}}(X_i, Y_i)\} \right] \\
&= \mathbb{E}_{D_+}\left[ \mathbb{E}_{E_+}\left[ \left( \max_{j \in [n+1]} |w(X_j, Y_j)| \right) \frac{1}{n+1} \sum_{i \in [n+1]} \mathbb{1}\{\widetilde{S}_i = \widehat{q}_{Y_{n+1}}(X_i, Y_i)\} \,\bigg|\, \{\Phi(X_i, Y_i)\}_{i \in [n+1]} \right] \right] \\
&= \mathbb{E}_{D_+}\left[ \left( \max_{j \in [n+1]} |w(X_j, Y_j)| \right) \frac{1}{n+1} \mathbb{E}_{E_+}\left[ \sum_{i \in [n+1]} \mathbb{1}\{\widetilde{S}_i = \widehat{q}_{Y_{n+1}}(X_i, Y_i)\} \,\bigg|\, \{\Phi(X_i, Y_i)\}_{i \in [n+1]} \right] \right]
\end{aligned}
$$

We will now bound the inner expectation of the above expression by showing that conditioning on $\{\Phi(X_i, Y_i)\}_{i \in [n+1]}$

$$
\mathbb{P}\left[ \sum_{i \in [n+1]} \mathbb{1}\{\widetilde{S}_i = \widehat{q}_{Y_{n+1}}(X_i, Y_i)\} > d \,\bigg|\, \{\Phi(X_i, Y_i)\}_{i \in [n+1]} \right] = 0.
$$

In more detail, we can upper bound this probability as follows

$$
\begin{aligned}
&\mathbb{P}\left[ \sum_{i \in [n+1]} \mathbb{1}\{\widetilde{S}_i = \widehat{q}_{Y_{n+1}}(X_i, Y_i)\} > d \,\bigg|\, \{\Phi(X_i, Y_i)\}_{i \in [n+1]} \right] = \\
&\mathbb{P}\left[ \exists 1 \leq j_1 < \ldots < j_{d+1} \leq n+1 \text{ s.t. } \forall i \in [d+1], \widetilde{S}_i = \widehat{q}_{Y_{n+1}}(X_i, Y_i) | \{\Phi(X_i, Y_i)\}_{i \in [n+1]} \right] \leq \\
&\sum_{1 \leq j_1 < \ldots < j_{d+1} \leq n+1} \mathbb{P}\left[ \forall i \in [d+1], \widetilde{S}_i = \widehat{q}_{Y_{n+1}}(X_i, Y_i) | \{\Phi(X_i, Y_i)\}_{i \in [n+1]} \right] \leq \\
&\sum_{1 \leq j_1 < \ldots < j_{d+1} \leq n+1} \mathbb{P}\left[ \exists w \in \mathcal{W} \text{ s.t. } \forall i \in [d+1], \widetilde{S}_i = w(X_i, Y_i) | \{\Phi(X_i, Y_i)\}_{i \in [n+1]} \right] = \\
&\sum_{1 \leq j_1 < \ldots < j_{d+1} \leq n+1} \mathbb{P}\left[ \exists \beta \in \mathbb{R}^d \text{ s.t. } \forall i \in [d+1], \widetilde{S}_i = \sum_{k \in [d]} \beta_k \Phi_k(X_i, Y_i) \,\bigg|\, \{\Phi(X_i, Y_i)\}_{i \in [n+1]} \right] = \\
&\sum_{1 \leq j_1 < \ldots < j_{d+1} \leq n+1} \mathbb{P}[(\widetilde{S}_{j_1}, \ldots, \widetilde{S}_{j_{d+1}}) \in \text{RowSpace}([\Phi(X_{j_1}, Y_{j_1})| \ldots |\Phi(X_{j_{d+1}}, Y_{j_{d+1}})]) | \{\Phi(X_i, Y_i)\}_{i \in [n+1]}].
\end{aligned}
$$

We notice that $R\left([\Phi(X_{j_1}, Y_{j_1})| \ldots |\Phi(X_{j_{d+1}}, Y_{j_{d+1}})]\right)$ is a $d$-dimensional subspace of $\mathbb{R}^{d+1}$. Since for fixed $\{\Phi(X_i, Y_i)\}_{i \in [n+1]}$ the scores $\widetilde{S}_{j_1}, \ldots, \widetilde{S}_{j_{d+1}}$ are independent and continuously distributed, we have that for all $1 \leq j_1 < \ldots < j_{d+1} \leq n+1$

$$
\mathbb{P}\left[ \left( \widetilde{S}_{j_1}, \ldots, \widetilde{S}_{j_{d+1}} \right) \in R\left([\Phi(X_{j_1}, Y_{j_1})| \ldots |\Phi(X_{j_{d+1}}, Y_{j_{d+1}})]\right) | \{\Phi(X_i, Y_i)\}_{i \in [n+1]} \right] = 0.
$$

Combining the inequalities of the steps above, we have proven that for all $w \in \mathcal{W}$

$$
|\mathbb{E}_{D,E}[\text{wCD}(\mathcal{C}, \alpha, w)]| \leq \frac{d}{n+1} \mathbb{E}_{D_+}\left[ \max_{i \in [n+1]} |w(X_i, Y_i)| \right].
$$

$\square$

# C. Additional Experimental Details

We use Histogram-based Gradient Boosting Tree through the implementation of scikit-learn (Pedregosa et al., 2011). Specifically, we use the HistGradientBoostingClassifier to train the basis weight functions for Kandinsky conformal prediction. We use HistGradientBoostingRegressor to train the base model for ACSIncome. We apply default hyperparameters suggested by scikit-learn except that we set max_iter to 250.

## C.1. ACSIncome

We preprocess the dataset following Liu et al. (2023). We additionally apply logarithmic transformation of labels with base 10 and scale the label to $[0, 1]$ by min-max scaling.

We train the base Gradient Boosting Tree regressor on 31,000 samples with 10,000 from each state. The calibration set contains 4,000 samples per state and the test set contains 2,000 samples per state. Given a number of groups $|\mathcal{G}|$, we select samples from states with the $|\mathcal{G}|$ smallest indices. We train the basis of Kandinsky's weight function class on the training set with selected states, but the base predictor is not retrained. We also filter the calibration and test set for selected states. We repeat the experiments 100 times by reshuffling the calibration and test set, but the training set is fixed such that Kandinsky's weight function class is also fixed for a given number of selected states.

We take the deterministic score function of Conformalized Quantile Regression (CQR). Given base predictors $f_{\alpha/2}(x)$ and $f_{1-\alpha/2}(x)$ for the $\alpha/2$ and $1 - \alpha/2$ quantile, respectively,

$$S(x, y) = \max \left\{ y - f_{1-\alpha/2}(x), f_{\alpha/2}(x) - y \right\}, \quad \widetilde{S} = S.$$

## C.2. CivilComments

Following Koh et al. (2021), we split the dataset into 269,038 training samples and 178,962 samples for calibration and test. We finetune a DistilBERT-base-uncased model with a classification head on the training set, following the configurations of Koh et al. (2021). We randomly redistribute samples between the calibration and test sets to vary the calibration sample sizes. Since the groups are overlapping, we estimate the ratio between the group average sample size and the overall sample number of 178,962. Then we downsample the dataset accordingly to approximately reach a prescribed group average sample size in the calibration set. We repeat the redistribution procedure 100 times, but the training set is fixed such that the DistilBert model is trained only once. However, we train the weight functions of Kandinsky conformal prediction on the calibration set, since we are considering the setup where training samples are only accessible to algorithms by the base predictor. Therefore, the weight function class is retrained for each calibration set. Since group sample sizes can be different between runs, in Figures 4(a) and 4(b), annotated group sample sizes are estimated by the mean of actual group sample sizes over 100 runs.

We take the randomized score function of Adaptive Prediction Sets (APS). Given the base classifier $f(x)$ that outputs the probability vector for all classes, we sort their probabilities in decreasing order.

$$f_{(1)}(x) \geq f_{(2)}(x) \geq ... \geq f_{|\mathcal{Y}|}(x).$$

We use $f_y(x)$ to represent the component of $f(x)$ for the class $y$ and $k_x(y)$ to represent the order of $f_y(x)$ such that $f_{(k_x(y))}(x) = f_y(x)$. The non-conformity score is given by

$$\widetilde{S}(x, y, \varepsilon) = \sum_{k=1}^{k_x(y)-1} f_{(k)}(x) + \varepsilon f_y(x), \quad \varepsilon \sim \text{Uniform}[0, 1].$$

