# OpenReview forum: "Kandinsky Conformal Prediction: Beyond Class- and Covariate-Conditional Coverage"
_ICML.cc/2025/Conference — ICML 2025 poster_

### Official Review · Reviewer_mFd6 · 2025-03-08

**Overall Recommendation:** 3

**Summary:**

The present paper introduces an approach to weighted conformal prediction that employs quantile regression as a solution. The authors study general weights depending both on covariates and labels and provide theoretical guarantees via high probability upper bounds. Several particular examples of weighting functions are considered as corollaries of general results. Online setup is also considered separately. Some numerical illustrations are provided.

## Update after rebuttal

I appreciate the review by authors. I think that the paper is generally sound and, while not having a single big contribution, improves the existing results in mainly generality of approach and sharpness of theory. I was slightly confused by the rebuttal by authors where at several places they were overclaiming the contributions (which is actually much less present in the paper itself). In particular, there is no new algorithm in the paper (adding Y as input doesn't lead to new algorithm). I increase my score by 1 point to acknowledge my general positive impression about the paper. However, I recommend the authors to further improve the clarity of the paper. For example, the introduction of randomized score is not adequately done in the paper.

**Claims And Evidence:**

There are 3 main claims in the paper:
1. First study of the general formulation of the (group) weighted conformal prediction which considers weights depending both on labels and covariates.

While, to the best of my understanding, previous work indeed didn't consider the weights dependent on labels in the context of group-conditional coverage, it is not clear whether this significantly changes the methods and and the theoretical analysis. Thus, I currently don't see whether this claim, while being formally correct, represents a substantial novelty with respect to existing literature.

2. Introduction of the new method for conformal inference.

The introduced method coincides with already existing methods except for using labels as input variables for quantile regression. I don't think that calling the method a new one is warranted.

3. Experimental evaluation of the proposed method

The experimental evaluation is very lightweight with just two datasets considered. For more theoretical paper, I don't necessarily see it as a drawback but also don't think that this part of the paper represents a strong contribution.

**Essential References Not Discussed:**

The authors generally relate the paper well to the existing literature on conformal prediction with distribution shifts. However, they claim that weights depending both on X and Y were not considered in the literature. However, it is not correct as the paper [1] considers general weights. Also, that paper performs the analysis generally similar to the one performed in the present paper except that it does not consider a VC class for the weight function.

[1] Plassier, Vincent, et al. "Efficient conformal prediction under data heterogeneity." International Conference on Artificial Intelligence and Statistics. PMLR, 2024.

**Experimental Designs Or Analyses:**

I think that the experiments are generally sound though only two datasets are considered. I am somewhat confused with the choice of baselines as I don't see which of the methods uses only covariates for quantile regression (without using the labels).

**Methods And Evaluation Criteria:**

The methodological part of the paper looks correct.

**Other Comments Or Suggestions:**

I think that the additional major issue with the paper is writing which is while being generally clear, fails to deliver by what means the paper achieves the results that are claimed. It includes the explanation of algorithmic and theoretical advances which are not properly explained (or their novelty is not explained to be precise). One of the particularly odd issues with writing is absence of any discussion of the score function. In particular, the authors introduce randomized score function $\tilde{S}$ but failed to find in the paper any discussion of this function or example. As one more example, Section 4 introduces test-time algorithm and discusses its drawbacks (computational complexity). However, it doesn't discuss any benefits of this procedure which makes the reader confused.

**Other Strengths And Weaknesses:**

I generally keep up with the literature in this area though I was not aware of the couple of recent (2024) papers on the topic that are referenced by authors.

**Questions For Authors:**

1. What are the algorithmic innovations of the paper (if any)?

2. What are the difficulties in theoretical analysis (if any) which one needs to overcome considering $w(X, Y)$ compared to simpler case of $w(X)$?

3. Why you obtain the rate which us better by $d^{1/2}$ than previous works?

4. What are the benefits of considering test-time algorithm?

**Relation To Broader Scientific Literature:**

I think that the paper generally well relates to the existing literature mentioning the majority the relevant papers. However, the corresponding discussion is not complete as the authors do not elaborate on why and how the paper succeeds to generalize the previous works. More precisely, it is not clear why considering $w(X, Y)$ is more challenging than considering just $w(X)$.

**Theoretical Claims:**

I didn't check the proofs line by line but the results well correspond to the existing literature and look sound. I guess an improvement of $d^{1/2}$ in one of the bounds is due to explicit consideration of finite dimensional weight function (as opposed to the general VC class) but I might be wrong.

---

> ### Author Rebuttal · Authors · 2025-04-01
>
> We thank you for your review. Below, we address your questions:
>
> **Q1**. Most conformal prediction algorithms share a common structure, involving quantile regression followed by constructing prediction sets via comparison with a quantile threshold. However, significant differences emerge in the precise choice of the quantile function class and the selection of samples used for regression—these subtle yet critical choices underpin our algorithmic contributions.
>
> *Weight functions jointly dependent on covariates and labels*: Although inspired by Gibbs et al. (2023), we generalize their linear quantile regression approach from weight functions defined solely on covariates to those defined jointly on covariates and labels. This generalization is key to achieving broader and practically relevant coverage guarantees (e.g., overlapping and latent groups).
>
> *Efficient quantile regression without monotonicity assumptions*: While our Alg. 3 directly extends Gibbs et al. (2023), ensuring computational efficiency is non-trivial due to the loss of monotonicity structure utilized in their method. This monotonicity structure no longer holds when we consider groups jointly defined by $X$ and $Y$. To address this, our main method, Alg. 1, extends the computationally efficient quantile regression technique from Jung et al. (2023)—originally limited to covariate-conditional guarantees—to handle weighted coverage guarantees. This algorithm avoids the need to solve a quantile regression problem for all candidates y at test time.
>
> *Generalization of Mondrian prediction sets*: Alg. 2 further adapts the classic Mondrian conformal prediction approach to our generalized weighted setting.
>
> **Q2**. The primary challenge arises because both the conformity score and the quantile $\hat{q}$ depend on both $X$ and $Y$. For example, in the analysis of quantile regression we want to bound the number of calibration samples $(X_i, Y_i)$ for which the score equals the quantile. In general, when this number is small, you can get a more accurate quantile estimator. As in prior work, we do this by conditioning on the random variables that fix the quantile’s value. However, in our case, if the score function is deterministic, then fixing the values of $X$ and $Y$ fully determines the score (as opposed to Gibbs et al. (2023) where they only need to condition on $X$). To address this, we show that it suffices to condition on $\Phi (Χ, Y)$, which preserves uncertainty in the score. For non-continuous $S|\Phi$, we can consider any randomized score function $\tilde{S}$ that satisfies the continuity assumption.
>
> Another challenge arises when given a quantile function $\hat{q}$ and the features of a test point $X$, we want to compute the prediction set. The issue is that both the score function and the quantile depend on the value of $X$ and $y$. Inspired by Mondrian conformal prediction, we obtain the prediction sets by returning all the labels y where the score $S(X,y) \leq \hat{q}(X,y)$.
>
> **Q3**. Areces et al. (2024) prove uniform convergence of the weighted coverage for the entire function class. We sharpen the dependence on $d$ by focusing on the basis weight functions, since the function class is a vector space.
>
> **Q4**. The alternative test-time algorithm in Sec. 4 achieves a different type of coverage guarantee based on exchangeability arguments. Specifically, this guarantee incorporates randomness over both the calibration dataset and the test point. In contrast, Alg. 1 and 2 provide a training-conditional guarantee, conditioning only on the fixed realization of the calibration data. Although these two types of guarantees are not directly comparable, exchangeability-based guarantees generally yield smaller coverage error bounds, as in our results in Sec. 4.
>
> **“the paper [1] considers … weight function.”**
>
> Thank you for bringing [1] to our attention. Both this paper and ours address potential distribution shifts between the calibration and the test data, where the distribution over the features and the label changes. The main difference between the two papers is that we get coverage guarantees for a class of distribution shifts, while they consider a single distribution shift that they can estimate using samples from the test distribution.
>
> **“particularly odd issues … discussion of the score function.”**
>
> In Sec. 3 we mention that “As is common in conformal prediction, our method ensures the desirable coverage for any score function $S$ [that satisfies the continuity assumption]”. For deterministic scores violating the continuity assumption, we mention that “we allow for the use of a randomized score function to break potential ties in quantile regression while keeping the assumptions about distribution $D$ minimal”. In our experiments we use the deterministic Conformalized Quantile Regression score function (Romano et al., 2019), and the randomized Adaptive Prediction Sets (APS) score function (Romano et al., 2019; Ding et al.,2023).

---

> > ### Comment · Reviewer_mFd6 · 2025-04-05
> >
> > I appreciate the answer by reviewers. I have several additional questions:
> > 1. "Alg. 1, extends the computationally efficient quantile regression technique from Jung et al. (2023)—originally limited to covariate-conditional guarantees—to handle weighted coverage guarantees". Algorithm 1 is a standard quantile regression formulation. Essentially, it is even not an algorithm as no procedure for solving the problem is provided. What do you mean precisely?
> >
> > 2. "Alg. 2 further adapts the classic Mondrian conformal prediction approach to our generalized weighted setting." The algorithm two is just comparison the score value with the threshold. What kind of adaptation do you mean?
> >
> > 3. Your answer on Q2: can you point to the line(s) in the proof(s) which execute(s) this conditioning?
> >
> > 4. Regarding the randomness in the score function what I mean that it appears in the paper from nowhere. There are no examples, no introduction of $\epsilon_i$, no discussion why is it needed.

---

> > > ### Author Response · Authors · 2025-04-05
> > >
> > > Thank you for your reply. To answer your questions:
> > >
> > > 1. You are correct that our Alg.1 takes the same form of linear quantile regression as Jung et al. (2023). However, a key difference lies in the function class over which we perform quantile regression. In comparison, Jung et al. fit quantile regressors over features of the form $\phi(x)$ (i.e., functions of covariates alone), but we perform quantile regression over a richer feature class $\phi(x, y)$, which depends jointly on both covariates and labels. This change requires a new way to compute the prediction set test time (our Alg. 2). We agree that our previous response could be clearer—a more precise statement would be that our combination of Alg. 1 and Alg. 2 extends the approach of Jung et al.
> > > We will explain how Alg. 2 is different in the answer to your next question.
> > >
> > > 2. The key distinction with Mondrian conformal prediction is that our method applies a per-example threshold for each candidate label $y$, based on the pair $(X_{n+1}, y)$. This threshold is obtained by evaluating a quantile regression model at $\phi(X_{n+1}, y)$. In contrast, Mondrian conformal prediction applies a per-group threshold: it first assigns $(X_{n+1}, y)$ to a discrete group, then uses a fixed threshold associated with that group. As a result, Mondrian conformal prediction requires the groups to be disjoint.
> > > In addition, while Jung et al. (2023) can handle overlapping groups, their per-example threshold function is only a quantile regression function of the covariates $X_{n+1}$. As a result, they can apply the same threshold for all candidate labels y.
> > >
> > > 3. In the proof, the conditioning on $\Phi$ occurs in L1004-L1038.
> > >
> > > 4. Thank you for clarifying your question. In the paper, we did formally introduce the noise $\varepsilon_i$ (left column of L176-181), an example of randomized Adaptive Prediction Sets (APS) score function (right column of L 356-357; detailed description in L 1090), and a discussion of why randomization is needed (left column L 173-176; in addition, a description of how it shows up in our theorem statement in the paragraph of left column L 219). To re-state our discussion here: “... the use of a randomized score function to break potential ties in quantile regression, while keeping the assumptions about distribution D minimal.”
> > > We are happy to expand the discussion to convey the following point: our method ensures the desirable coverage for any score function that satisfies the continuity assumption that $S|\Phi$ is continuous. For deterministic scores violating the continuity assumption, we need to randomize the score.

---

### Official Review · Reviewer_1N7z · 2025-03-14

**Overall Recommendation:** 4

**Summary:**

The paper proposes Kandinsky Conformal Prediction method for general group-conditional guarantees in contrast to Mondrian conformal prediction that ensures conditional coverage over a disjoint set of groups. Their framework handles overlapping and fractional group memberships, and allows for group memberships to be jointly defined by covariates and labels. They show this method is computationally efficient as it requires solving a single quantile regression over the vector space spanned by group functions; and statistically optimal as it matches the minimax-optimal conditional coverage error rate with finite samples. Through empirical evaluation on ACSIncome and CivilComments datasets, authors demonstrate their method achieves the best conditional coverage as measured by best group average coverage deviation and stable average coverage deviation with the increase in number of groups.

**Claims And Evidence:**

The claims made in the submission regarding their general formulation of conditional coverage are supported by their theory (Theorem 3.1, Corollary 3.4) and accompanying proofs. The experiments empirically validate these claims.

**Essential References Not Discussed:**

I feel the paper is fairly complete in its discussion of important references for understanding the context.

**Experimental Designs Or Analyses:**

I checked the experimental analysis for both ACSIncome and CivilComments datasets. I do not see any issues with the current design choices, but there is definitely scope to make the empirical evaluation more extensive (in terms of datasets, score functions, as well as choice of the function class).

**Methods And Evaluation Criteria:**

Yes, the proposed method provides a general framework for conditional conformal prediction, where the weight function class can be adapted to obtain group-conditional guarantees, Mondrian conformal prediction, conformal prediction with fractional group membership, and distribution shifts. The evaluation metric of group average coverage deviation also makes sense.

**Other Comments Or Suggestions:**

I would like to see more details regarding the implementation of the algorithm (e.g., expanding section A.3), especially as the code was not shared in the current submission.

**Other Strengths And Weaknesses:**

I appreciate the general framework proposed and the flexibility it offers to obtain the desired coverage guarantees for several applications. I also found the paper to be well-written and the proofs very detailed.

Refer to questions below for specific concerns regarding empirical evaluation.

**Questions For Authors:**

Is there any reason why comparison with Gibbs et al., 2023 and Ding et al., 2023 (class conditional conformal prediction) is not included in the experiments? I believe this would help understand the performance improvements offered by the proposed method in the very specific cases introduced in the paper.

**Relation To Broader Scientific Literature:**

This paper expands the existing conditional coverage guarantees to allow overlapping groups and fractional memberships that are defined over both covariates and labels. As the paper mentions, when weight functions are based on covariates, the method provides the same type of guarantees as in past work but with tighter bounds on expected weighted coverage deviation.

**Theoretical Claims:**

I went over the proofs for Theorem 3.1 and Corollary 3.4 in the Appendix.

---

> ### Author Rebuttal · Authors · 2025-04-01
>
> We would like to thank you for your review. We address your question and your comment below.
>
> ### 1. I would like to see more details regarding the implementation of the algorithm (e.g., expanding section A.3), especially as the code was not shared in the current submission.
>
> This links to the code: https://doi.org/10.5281/zenodo.15104271. We will include the code in the publication and we are ready to share further implementation details.
>
> ### 2. Is there any reason why comparison with Gibbs et al., 2023 and Ding et al., 2023 (class conditional conformal prediction) is not included in the experiments? I believe this would help understand the performance improvements offered by the proposed method in the very specific cases introduced in the paper.
>
> We have compared Kandinsky conformal prediction with class conditional conformal prediction with the CivilComments dataset. However, the class conditional method is not implementable for the regression task in ACSIncome. The method of Gibbs et al., 2023 does not scale for the image tasks we consider because it trains the image classifier from scratch to learn the weight functions. This violates the post-hoc nature of conformal prediction. Kandinsky trains the group classifier using the two-dimensional input of logits and labels, which is scalable.

---

> > ### Comment · Reviewer_1N7z · 2025-04-03
> >
> > 1. Thanks for sharing the code. Please expand the final paper to add further implementation details. If you can share the details in your response, that would also be helpful.
> >
> > 2. Is the class-conditional method Mondrian CP or clustered CP from Ding et al.? This detail is missing I suppose, and it would be helpful to add both as the performance depends on the specific regime as discussed in Ding et al. Also, I do not agree that Gibbs et al. method will not scale and that it *trains the image classifier from scratch*. It updates the quantile fit at test time -- while it is expensive to update the fit for each test point, the implementation shared by the authors uses a small number of iterations in practice. The experiments in their paper for image classification as well as experiments I have seen on datasets of larger scale validate that the experiments are feasible. Please consider updating your results in light of this or feel free to clarify anything that is missing. Also, which image tasks are you referring to -- I can only see experiments on ACSIncome and CivilComments.
> >
> > Overall, I still believe the empirical evaluation is limited in demonstrating the benefits of the method.

---

> > > ### Author Response · Authors · 2025-04-09
> > >
> > > We would like to thank you for your response! To answer your remaining questions:
> > >
> > > ### 1. Is the class-conditional method Mondrian CP or clustered CP from Ding et al.?
> > >
> > > We run class conditional conformal prediction on CivilComments, which has two classes.  In this setting, Mondrian CP and clustered CP from Ding et al. yield the same algorithm. This is because clustered CP treats each class as a distinct cluster.
> > >
> > > ### 2. Also, I do not agree that Gibbs et al. method will not scale and that it trains the image classifier from scratch. It updates the quantile fit at test time -- while it is expensive to update the fit for each test point, the implementation shared by the authors uses a small number of iterations in practice.
> > >
> > > We are referring to a ​​language classifier​​ (not an image classifier) in our discussion. We apologize for the earlier typo. Regarding Gibbs et al.'s approach: While they update the model fit for each test point with a relatively small number of iterations, they still require training a ​​language classifier from scratch​​. Here the language classifier refers to the classifier used to predict group memberships, which forms the basis for constructing the weight function class.

---

### Official Review · Reviewer_mj6f · 2025-03-18

**Overall Recommendation:** 3

**Summary:**

The paper proposes an extension of the conditional coverage works of Jung et al. and Gibbs et al. to functions of covariates and labels, not just covariates as in these previous works. Experiments on ACSIncome and CivilComments datasets are included.

**Claims And Evidence:**

Claims are supported by the evidence in the paper.

**Essential References Not Discussed:**

See Blot et al. https://arxiv.org/abs/2406.17819 for an extension of the GCC paper to also include weight functions that depend on the label as well as the covariate. The algorithm in Section 4 should be subsumed by this paper, although the two-sided bound is new.

**Experimental Designs Or Analyses:**

Although the analysis looks correct, there is no code available to double check.

**Methods And Evaluation Criteria:**

Yes.

**Other Comments Or Suggestions:**

All the arguments in GCC'24 already hold for randomized score functions. There is no novelty in using a randomized score in the quantile regression.

**Other Strengths And Weaknesses:**

The paper provides the same flavor of bounds as GCC'24, but for weight functions that depend on both X and Y. Some initial work in this direction was completed by Blot et al (https://arxiv.org/abs/2406.17819), but this paper takes it yet another step further by proving a larger set of results including two-sided bounds.

It is a reasonable contribution to the field to extend the GCC'24 analysis to these , although it seems to include no highly novel technical devices. The experiments are reasonable for the purpose of demonstrating validity of these results, although not extraordinary.

**Questions For Authors:**

Unclear what randomness wCD is integrating over. if C is an argument, then it can be randomized (as in the case of a random score). Then in what sense does the inequality in Theorem 3.1 hold? Almost surely?

**Relation To Broader Scientific Literature:**

Extending the results of GCC'24 to test-label conditional coverage guarantees.

**Theoretical Claims:**

I did not carefully check the proofs.

---

> ### Author Rebuttal · Authors · 2025-04-01
>
> Thank you for your constructive feedback. We address your concerns below.
>
> ### 1. “The paper provides the same flavor of bounds as GCC'24, but for weight functions that depend on both X and Y. Some initial work in this direction was completed by Blot et al (https://arxiv.org/abs/2406.17819), but this paper takes it yet another step further by proving a larger set of results including two-sided bounds.”
>
> Thank you for pointing us to the paper by Blot et al. In our paper, we provide results that are stronger than Blot et al. in two ways. First, Algorithm 3 and the bound in Theorem 4.1 in our paper have a similar flavor to the results in Section 2.2 of Blot et al. However, their result is a one-sided bound, whereas we derive a two-sided bound assuming that the data are i.i.d. and that the distribution of $S(X,Y)|\Phi(X,Y)$ is continuous. Second, in Section 3 we obtain high probability bounds (that are in general stronger than expectation bounds) using Algorithms 1 and 2, under the same assumptions. We will incorporate this citation and discussion into our paper.
>
> ### 2. All the arguments in GCC'24 already hold for randomized score functions. There is no novelty in using a randomized score in the quantile regression.
>
> The use of randomized score functions is not the key contribution of our work. The main reasons why randomized score functions are used in Gibbs et al. (2023) and our paper though are somewhat different. In Gibbs et al. (2023) the authors use randomized score functions to obtain exact coverage. The difficulty when extending quantile regression to work for weight functions of the form $w(X,Y)$ is that conditioning on $X$ and $Y$, the value of a deterministic score function $S$ is fully defined (as opposed to the setting in Gibbs et al. (2023)). Therefore, we add randomization to the score function as one of the ways to overcome the obstacle of $S(X,Y)|X,Y$ not being continuous for a deterministic $S$.
>
> ### 3. Unclear what randomness wCD is integrating over. if C is an argument, then it can be randomized (as in the case of a random score). Then in what sense does the inequality in Theorem 3.1 hold? Almost surely?
>
> In Section 2 we define wCD as an expectation over the test point $(X,Y)$ and the internal randomness of $\mathcal{C}$ (which exists when as you mention we use a randomized score of the test point). Since wCD is by definition integrating over the randomness of $\mathcal{C}$, in Theorem 3.1 the dependence on the randomness of the score of the test point is accounted for in wCD. To explain things further, the guarantee in Theorem 3.1 holds with high probability over the randomness of the calibration dataset and the noise $\{\varepsilon_i\}_{i\in [n]}$ that is used to randomize the scores of the calibration data points.

---

### Official Review · Reviewer_EL5A · 2025-03-18

**Overall Recommendation:** 4

**Summary:**

This paper is considering the problem of conformal prediction with conditional guarantees, particular allowing the conditioning event to be both a function of covariate X and label Y. They build upon the previously proposed technique in the literature which is based on training a linear quantile regression over the linear span of some predefined basis. This basis would then capture a finitely many conditioning events, which in this paper can be a function of X and Y. They prove both PAC-type and "averaged over calibration"-type coverage guarantees and have a number of experiments backing up their claims.

**Claims And Evidence:**

I believe all the claims in the paper are accurate. They accompany their propose method with a number of special cases that makes everything crystal clear.

**Essential References Not Discussed:**

Not that I know of.

**Experimental Designs Or Analyses:**

They make sense to me.

**Methods And Evaluation Criteria:**

The evaluation methods make sense to me.

**Other Comments Or Suggestions:**

look at questions ...

**Other Strengths And Weaknesses:**

look at questions ...

**Questions For Authors:**

Even though the extra dependance on Y in designing the conditioning event makes the framework more flexible, however it highlights the question of how should we design the basis (\phi) in practice? This problem also applies to the works of [Gibbs et al] and [Jung et al] on covariate conditional events. Of course in some case we can design the groups relying on domain knowledge of fairness considerations and so on. But more generally, the question is how do we systematically figure out what kind of conditioning event we have to include or more generally what kind of distribution shifts we have to consider. Now for the case of covariate shifts, there is this option of taking advantage of unlabeled data from the test domain, which might be accessible in some scenarios. But unlabeled data would not be useful to figure out a good basis function beyond covariate shifts. So how can we design a meaningful basis as a function of X and Y?
In thinking about these it might also be good to take a look at some other research directions like [1] which are thinking about this question in the setting of covariate shift, but might give some ideas for the more general case in this paper.

[1] conformal prediction with learned features

**Relation To Broader Scientific Literature:**

They broaden the use case of conformal prediction in the scenarios where conditional coverage is important. The extra dependance on Y makes this framework more flexible from the application point of view as it can model more advanced subpopulations of the data or more advanced distribution shifts (beyond covariate shift).

**Theoretical Claims:**

All the theoretical claims make sense. I have looked through their proofs, the proofs are very similar to the work of [Gibbs et al 2024], which is based on analyzing the sub gradients of the quantile regression. The resemblance is natural as they are generalizing their method. That being said, it might be the case that I have missed some small errors, as I did not check all the equations one by one.

---

> ### Author Rebuttal · Authors · 2025-04-01
>
> We would like to thank you for your review. We answer your question below.
>
> ### 1. “... how should we design the basis (\phi) in practice? … how do we systematically figure out what kind of conditioning event we have to include or more generally what kind of distribution shifts we have to consider … How can we design a meaningful basis as a function of X and Y?”
>
> This is an interesting research direction. Towards this direction, we analyze theoretically and evaluate empirically the application of fractional group membership where we obtain the desired coverage for groups defined by unobserved attributes. For this application in Section 3.1 we define $\Phi(X,Y)$ as the probability that the test point $(X,Y)$ belongs to a group $G$ based on the unobserved attributes conditioned on the value of $(X,Y)$. Our theoretical analysis is for the case where we know the true probabilities that define $\Phi(Χ,Υ)$. In practice, we can use the calibration data that includes the value of the protected attribute to estimate the probabilities for the basis $\Phi$. We implement this approach in our experiments in Section 5.

---

### Official Review · Reviewer_QKpU · 2025-03-20

**Overall Recommendation:** 3

**Summary:**

The authors build on the rich literature on extensions of the conformal prediction framework, that traditionally yields intervals where marginal probabilistic calibration is true.
More specifically, they propose a method able to infuse very generic notions of group-conditional calibration, moving from previous work covering only more specific cases.
Such method, called Kandinsky Conformal Prediction, is thus proposed, described in detail and then tested on real world tasks.

**Claims And Evidence:**

From a theoretical perspective, Kandinsky CP is fully charachterised by an extensive theoretical analysis, with well proven results.
I am a bit skeptical about the applicative case study, but more on it below.

**Essential References Not Discussed:**

All the fundamental literature pieces are there, I would only cite the second edition of Algorithmic Learning in a Random World (same authors, but published in 2023)

**Experimental Designs Or Analyses:**

While I find the theoretical endeavour of sure and clear merit, I remain a bit skeptical about the applicative case studies.
Their results show that Kandinsky CP gives better results than the methodologies already available in the literature.
I believe the paper is lacking in this respect along two directions

-It is unclear why the proposed method performs better, and thus when such an increase in complication is worth from an applicative perspective.
This can be achieved via a more extensive study on real world datasets, in order to charachterise the types of analytical scenarios where Kandinsky beats other methods
-Along this same directions, it would be interesting to charachterise the performance of the method also via the use of synthetic datasets... e.g. I would be interesting to know what degree of overlap between classes would make Kandinsky better than the alternative.

**Methods And Evaluation Criteria:**

The applicative benchmarks are well chosen and very representative (maybe a bit limited).

**Other Comments Or Suggestions:**

none of relevance

**Other Strengths And Weaknesses:**

none of relevance

**Questions For Authors:**

Your test examples seem to show a supremacy of Kandinsky CP with respect to other methodologies. Do you have any idea why? Is it because of specific features of the dataset, or just because the method is better than the alternatives allround?

Are there applicative situations where Kandinsky CP "breaks down"?

This can be explored also with synthetic datasets.

**Relation To Broader Scientific Literature:**

As stated previously, this result is generated in the context of the rich body of literature about non-marginal conformal prediction. The authors make a good effort at summarising the fundamental references.

**Theoretical Claims:**

I have checked all the proofs in the manuscript, and they seem correct.

---

> ### Author Rebuttal · Authors · 2025-04-01
>
> We would like to thank you for your valuable comments. We address your questions and concerns below.
>
> ### 1. “It is unclear why the proposed method performs better…Kandinsky beats other methods.” “Your test examples seem to show… alternatives allround?”
>
> We provide two concrete example scenarios where KCP's generality directly improves coverage:
>
> - **Overlapping Groups:** We consider overlapping subgroups jointly defined by covariates $X$ and labels $Y$. For example, in our experiments with the ACSIncome dataset (income prediction task), groups are naturally overlapping and jointly defined by subsets of demographic attributes and income levels. In such scenarios, standard Mondrian conformal prediction does not apply, as it requires disjoint groups. Furthermore, the conservative baseline method introduced by Barber et al. (2021)—which assigns the most conservative prediction sets to test points based on all overlapping groups to which they belong—tends to produce excessively large and thus impractical prediction sets. Our experiments also validate this result.
>
>
> - **Latent or Implicit Groups:** KCP effectively handles scenarios involving latent groups or implicit subpopulations defined by attributes unavailable at test time. Previous methods that rely exclusively on explicit group annotations are inadequate for such cases. For example, class-conditional conformal prediction uses only labels $Y$ as explicit group identifiers, making it insufficient for capturing latent groups. In contrast, the Kandinsky framework jointly leverages covariates $X$ and labels $Y$ to infer implicit, probabilistic group memberships. Our experiments with the CivilComments dataset precisely demonstrate this scenario: group annotations are unavailable at test time, and our results show that KCP successfully maintains the desired coverage guarantees, whereas baseline approaches like class-conditional CP fail to do so.
>
> Additionally, we want to emphasize that KCP unifies existing methods (class-conditional, covariate-group-conditional, Mondrian) without increasing complication. It reduces general group-conditional coverage to a single quantile regression over an appropriately chosen function class. Our main contribution rigorously shows that this approach achieves stronger conditional coverage guarantees through a practical and simple algorithm.
>
>
>
> ### 2. “I would be interested to know what degree of overlap between classes would make Kandinsky better than the alternative. “
> In the CivilComments test dataset, the number of groups that a sample belongs to follows the distribution below:
> | Number of Groups | Samples |
> |-----------|---------------|
> | 0         | 81235         |
> | 1         | 37226         |
> | 2         | 12041         |
> | 3         | 2704          |
> | 4         | 470           |
> | 5         | 88            |
> | 6         | 13            |
> | 7         | 4             |
> | 8         | 1             |
> 11% samples belong to at least two overlapping groups. Under such a degree of overlapping, Kandinsky achieves half the coverage deviation of the best alternative (the conservative conformal prediction method).
>
>
> ### 3. Are there applicative situations where Kandinsky CP "breaks down"?
>
> We conducted experiments to systematically analyze scenarios where Kandinsky Conformal Prediction (KCP) provides an advantage. Specifically, we explored the effects of varying two critical parameters: the number of groups and the sample size per group. In the ACSIncome dataset, we fixed the number of samples per group while progressively increasing the total number of groups. In the CivilComments dataset, we varied the total sample size while keeping the number of groups constant.
>
> Our results show that KCP scales robustly with an increasing number of groups, whereas baseline methods deteriorate significantly (Fig 1a). Additionally, KCP’s performance consistently improves with larger sample sizes. However, we observed a scenario where KCP underperforms compared to class-conditional conformal prediction when the sample size per group falls below approximately 250 in the CivilComments dataset (Fig 1d). This occurs because class-conditional prediction assumes binary groups defined solely by labels Y, giving it a larger "effective" sample size per group—up to eight times more than KCP, which handles 16 overlapping groups jointly defined by X and Y. Consequently, class-conditional prediction may initially appear competitive at small sample sizes, but its inherent limitations become clear as sample sizes grow. In summary, KCP demonstrates substantial practical advantages, particularly in settings with many groups and moderate to large sample sizes per group.

---

> > ### Comment · Reviewer_QKpU · 2025-04-02
> >
> > I thank the authors for the insightful answers, which I believe clarify the results of the paper.
> > I remain not fully satisfied though when it comes to the exploration of the benefits of a Kandinsky method wrt to a more classical one as a function of class overlap.
> > A theoretical study, or a more extensive theoretical one would have been very interesting from a methodological perspective.

---

### Official Review · Reviewer_oFZY · 2025-03-25

**Overall Recommendation:** 4

**Summary:**

This paper proposes a new conditional conformal prediction framework named **Kandinsky conformal prediction** , which considers the conditional coverage given the information from both covariate and label. The theoretical results also improved existing bounds.

**Claims And Evidence:**

Yes.

**Essential References Not Discussed:**

No.

**Experimental Designs Or Analyses:**

Yes. The experiments are solid.

**Methods And Evaluation Criteria:**

Yes.

**Other Comments Or Suggestions:**

1. Add explicit definition of $C(X_{n+1})$ in Algorithm 3.

2. Provide examples or experiments for Fractional group memberships.

3. Discuss the technical reason for improving the convergence rate over  Roth (2022); Jung et al. (2023), and the sharper dependence on $|\mathcal{G}|$ compared with Areces et al. (2024).

**Other Strengths And Weaknesses:**

**Weaknesses**

1. The applications in Section 3 are not fully explored in experiments.

2. Lack of comparison results with Romano et al. (2020a).

**Questions For Authors:**

No.

**Relation To Broader Scientific Literature:**

The proposed method expands the scope of condition-conformal methods in Jung et al. (2023) and Gibbs et al. (2023).

**Theoretical Claims:**

I have checked the proofs in Section A.1, which are correct to me.

---

> ### Author Rebuttal · Authors · 2025-04-01
>
> We would like to thank you for your feedback. We address your comments below.
>
> ### 1. “Add explicit definition of C(X_{n+1}) in Algorithm 3.”
>
> The definition of $C(X_{n+1})$ is given in the equation of the second line of Algorithm 3, where it includes all $y$ such that the score function on the test point is smaller than a threshold $\hat q_y$. The threshold $\hat q_y$ is defined in the next equation, which is computed by quantile regression on both calibration samples and test samples.
>
> ### 2. “Provide examples or experiments for Fractional group memberships.”
>
> The ACSIncome experiment in the paper is designed with fractional group memberships. The dataset is derived from US Census data collected from different states, which are considered as groups. The state attribute is omitted from the covariates and test point labels, leading to fractional group membership: as defined in L220, we consider fractional group membership with groups defined by unobserved attributes, such that the membership can only be inferred by a probabilistic function over $X$ and $Y$. Given the ground-truth group, the membership is always deterministic. But the membership is fractional for the predictor which observes $X$ but does not observe the group.
> We propose another example in the introduction (L74). Due to regulatory restrictions, sensitive attributes are sometimes prevented from being used for prediction. Grouping by unobserved sensitive attributes causes fractional membership.
>
> ### 3. “Discuss the technical reason for improving the convergence rate over Roth (2022); Jung et al. (2023), and the sharper dependence on $|G|$ compared with Areces et al. (2024).”
>
> **Compared to Roth (2022); Jung et al. (2023):**
>
> We establish a connection between the ​subgradient of the empirical quantile loss and the ​empirical coverage guarantee, then prove concentration for the empirical coverage.
> Roth (2022) and Jung et al. (2023) connect the ​subgradient of the expected quantile loss to the ​expected coverage, so they must prove the concentration for the quantile loss itself. The empirical coverage yields tighter concentration because it is the sum of binary-valued functions. Roth (2022) and Jung et al. (2023) only prove concentration for a ​regularized quantile loss, and the regularization worsens sample complexity.
>
> **Compared to Areces et al. (2024):**
>
> Areces et al. (2024) prove uniform convergence of the weighted coverage for the whole function class, but we prove uniform convergence only for the basis weight functions since the function class is a vector space. This sharpens the dependence on |G|.
>
> ### 4. “Lack of comparison results with Romano et al. (2020a).”
>
> Romano et al. (2020a) consider a different setting from ours. They assume that the group label for the test point is known, while we address the more general case where the group label is latent. Therefore, their method cannot be directly applied in our experiments.

---

### Decision · Program_Chairs · 2025-05-01

**Decision:**

Accept (poster)

**Comment:**

The paper introduces Kandinsky Conformal Prediction (KCP), a novel framework for conformal prediction that extends conditional coverage guarantees to weight functions dependent on both covariates (X) and labels (Y). Six reviewers evaluated the submission, assigning overall recommendations of three Accept (score: 4) and three Weak Accept (score: 3). All reviewers commend the theoretical rigor and the paper’s contribution to generalizing conditional conformal prediction. During rebuttal, the reviewers' concerns are mainly on the limited empirical evaluation and clarity of the writing. The authors’ rebuttal adequately addresses most concerns, and the suggested improvements can be incorporated in the final version to strengthen the presentation. The consensus leans toward acceptance, with scores reflecting confidence in the work’s value to the conformal prediction community.